# SMCHD1 and LRIF1 converge at the FSHD-associated D4Z4 repeat and LRIF1 promoter yet display different modes of action

Darina Šikrová[1], Alessandra M. Testa [1,3], Iris Willemsen[1], Anita van den Heuvel[1], Stephen J. Tapscott[2], Lucia Daxinger[1], Judit Balog [1] & Silvère M. van der Maarel [1✉]

Facioscapulohumeral muscular dystrophy (FSHD) is caused by the epigenetic derepression of the 4q-linked D4Z4 macrosatellite repeat resulting in inappropriate expression of the D4Z4 repeat-encoded *DUX4* gene in skeletal muscle. In 5% of FSHD cases, D4Z4 chromatin relaxation is due to germline mutations in one of the chromatin modifiers SMCHD1, DNMT3B or LRIF1. The mechanism of SMCHD1- and LRIF1-mediated D4Z4 repression is not clear. We show that somatic loss-of-function of either SMCHD1 or LRIF1 does not result in D4Z4 chromatin changes and that SMCHD1 and LRIF1 form an auxiliary layer of D4Z4 repressive mechanisms. We uncover that SMCHD1, together with the long isoform of LRIF1, binds to the LRIF1 promoter and silences *LRIF1* expression. The interdependency of SMCHD1 and LRIF1 binding differs between D4Z4 and the *LRIF1* promoter, and both loci show different transcriptional responses to either early developmentally or somatically perturbed chromatin function of SMCHD1 and LRIF1.

[1]Department of Human Genetics, Leiden University Medical Center, 2333ZC Leiden, The Netherlands. [2] Human Biology Division, Fred Hutchinson Cancer Research Center, Seattle, WA 98109, USA. [3]Present address: Department of Biomedical Sciences, University of Padua, 35100 Padua, Italy. ✉email: S.M.van_der_Maarel@lumc.nl

Facioscapulohumeral muscular dystrophy (OMIM #158900 & #158901) is a heterogeneous disorder caused by mis-expression of the transcription factor DUX4 in skeletal muscle[1,2]. One of the key physiological roles of DUX4 is its involvement in zygotic genome activation at the human 4-cell cleavage stage[3,4]. A short burst of DUX4 expression during cleavage stage is followed by the activation of specific classes of retroelements and a cleavage stage-specific gene set. Indeed, this DUX4-sensitive transcriptional signature is also present in skeletal muscle biopsies and muscle cell cultures derived from individuals with FSHD or upon ectopic DUX4 expression in control myoblasts[5–7]. This and other evidence suggests that DUX4 is a pioneer transcription factor able to overwrite the existing chromatin environment in differentiated cell types and to activate its native transcriptional program[8–12]. DUX4 is encoded by a multicopy retrogene organized into the D4Z4 macrosatellite repeat located in the 4q and 10q subtelomeres[13]. While the exact origin of DUX4 expression at the cleavage stage has not been determined yet, typically only 4q D4Z4-derived DUX4 transcripts are associated with FSHD[1]. Furthermore, two major 4q subtelomeric allelic variants exist (4qA and 4qB), with only 4qA alleles being permissive for DUX4 expression in skeletal muscle tissue due to the existence of a polymorphic DUX4 polyadenylation signal[14], whereas this polyadenylation signal is absent on chromosome 4qB and 10q.

DUX4 expression is restricted to the 4-cell cleavage stage after which it is quickly attenuated[4], and the DUX4 locus remains transcriptionally silent in most somatic tissues[15,16]. In general, macrosatellite repeats in the genome, like D4Z4, adopt a heterochromatic structure in soma marked by high levels of DNA methylation and repressive histone modifications such as H3K9me3[17]. A partial failure in the establishment and/or maintenance of this epigenetic landscape at D4Z4 results in variegated DUX4 expression in FSHD myogenic cultures[1]. Successful D4Z4 repeat silencing is mainly dependent on the repeat copy number[18]. In the non-affected population, the D4Z4 repeat is polymorphic in size and consists of 8-100 repeat units. In FSHD individuals, two partially overlapping genetic mechanisms lead to a failure in epigenetic silencing of this locus, allowing for DUX4 expression in skeletal muscle and disease manifestation. In most FSHD cases (FSHD1; 95%), a contraction of the repeat to a size of 1–10 units on a 4qA allele occurs, which is associated with partial D4Z4 chromatin relaxation in somatic cells[19,20]. In the remaining 5% of FSHD cases (FSHD2), the epigenetic deregulation of D4Z4 results from trans mutations in chromatin factors that act on D4Z4[21–23]. In the latter case, the epigenetic landscape of both 4q and 10q D4Z4 repeats is affected, whereas in FSHD1 cases, only the contracted 4qA-D4Z4 repeat is epigenetically compromised[24,25]. The FSHD2 disease mechanism is also repeat size-dependent as mutations in D4Z4 chromatin modifiers only result in disease presentation when combined with a repeat size <20 D4Z4 units[26]. In fact, FSHD1 and FSHD2 seem to form a disease continuum with mutations in FSHD2 genes acting as disease modifier in upper-sized FSHD1 repeats[27].

Currently, mutations in three genes have been linked to FSHD: SMCHD1[21,28,29], DNMT3B[22] and LRIF1[23]. The most frequently mutated gene in FSHD2 is SMCHD1, accounting for >85% of FSHD2 individuals[21]. The SMCHD1 protein forms homodimers via its C-terminal hinge domain[30] and has a role in the topological organization of chromatin, predominantly at the murine inactive X chromosome[31–35]. Recent studies showed that maternal SMCHD1 levels affect the development of wild type embryos[36,37]. However, the mechanism by which SMCHD1 silences D4Z4 has not been fully answered yet. The somatic D4Z4 chromatin profile in FSHD2 cases with heterozygous SMCHD1 mutations is characterized by DNA hypomethylation, increased H3K4me2 levels and decreased H3K9me3 levels[23], similar to contracted D4Z4 repeats in FSHD1. In addition, increased levels of H3K27me3 are specifically found in FSHD2[38]. Several studies identified missense or nonsense mutations distributed over the entire SMCHD1 locus in FSHD2 but heterozygous missense mutations in the ATPase domain of SMCHD1 are also associated with the rare developmental disorder Bosma Arhinia and Microphtalmia Syndrome (BAMS; MIM603457)[39,40]. In these patients, SMCHD1 mutations also result in D4Z4 hypomethylation and DUX4 transcripts have been detected in some BAMS individuals[39–41].

The second gene identified as FSHD2 disease gene is DNMT3B. Heterozygous mutations in DNMT3B have been linked to FSHD2, while biallelic mutations in DNMT3B cause the Immunodeficiency, Centromeric instability, Facial anomalies type I (ICF1) syndrome (OMIM #242860)[42,43]. In both disease situations, 4q and 10q D4Z4 repeats are hypomethylated[22,44], and DUX4 expression has also been observed in ICF1 individuals with at least one 4qA allele, which puts them at risk for FSHD[22]. Similar to FSHD2 due to SMCHD1 mutations, the D4Z4 repeat in ICF1 individuals also shows reduced amounts of H3K9me3[45] and increased H3K4me2[16], while the status of H3K27me3 has not been investigated yet.

More recently, we have identified an individual presenting symptoms compatible with FSHD2 caused by a homozygous frameshift mutation in the LRIF1 gene combined with an 11 unit-long repeat on a 4qA chromosome. This homozygous frameshift mutation leads to the loss of the long LRIF1 isoform (LRIF1L), while the expression of the short isoform (LRIF1S) persists. The D4Z4 chromatin profile of the proband resembles that of FSHD2 individuals with heterozygous SMCHD1 mutations, including increased H3K27me3 levels consistent with the presence of DUX4 in myogenic cell cultures[23]. LRIF1 interacts with SMCHD1 via its coiled-coil C-terminal domain and is also enriched at the inactive X chromosome in somatic cells. Moreover, it interacts with all three HP1 paralogues via its PxVxL motif at its C terminus[46]. Maternal LRIF1 levels, like SMCHD1, regulate the development of wild-type embryos[37].

The emergence of the D4Z4 epigenetic abnormalities in FSHD1 and 2 is not well known. However, in the case of ICF1 and FSHD2 individuals with germline DNMT3B mutations, it is most likely during early developmental stages when DNMT3B establishes the cells' DNA methylation profiles. The time window and molecular activity of SMCHD1 and LRIF1 that enforces a repressive D4Z4 chromatin structure in somatic cells is less clear. On one hand, ectopic expression of SMCHD1 in FSHD myoblasts[38] and its mutation correction in FSHD2 myoblasts[47] results in DUX4 downregulation, suggesting that SMCHD1 does have a role in DUX4 repression also in somatic cells. However, these studies did not thoroughly examine the D4Z4 chromatin state after modulating SMCHD1 levels. On the other hand, a recent study showed that knocking out SMCHD1 in HCT116 colon carcinoma cells leads to DUX4 derepression without changes in DNA methylation or H3K9me3 levels at D4Z4 suggesting that SMCHD1 is not required for the maintenance of these epigenetic marks in this cell line[41]. In addition, transient knock-down of the long LRIF1 isoform results in DUX4 transcriptional derepression in control as well as in FSHD1 and FSHD2 myoblasts[23] suggesting that it too has a DUX4 expression modifying role in somatic cells albeit with unknown effect on the D4Z4 chromatin structure. Therefore, it is imperative to examine the role of SMCHD1 and LRIF1 in DUX4 repression in somatic cells as well as studying the chromatin requirements for their D4Z4 recruitment. Here, we examined SMCHD1 and LRIF1-mediated DUX4 repression in different somatic cell model systems with distinct D4Z4 chromatin environments and

demonstrate that they provide an auxiliary layer of chromatin repression on top of DNA methylation and H3K9me3. We also uncover an SMCHD1 and LRIF1-mediated transcriptional (auto) regulation of the *LRIF1* locus itself in somatic cells and show that this regulation differs from the action that SMCHD1 and LRIF1 impose on D4Z4 suggesting alternative modes of repression mediated by these two proteins at different genomic loci.

## Results

**Somatic loss of LRIF1 or SMCHD1 in control myocytes leads to insufficient *DUX4* derepression.** We have previously shown that SMCHD1 and LRIF1L aid in transcriptional repression of the D4Z4 repeat as short-term depletion of either LRIF1L or SMCHD1 in muscle cells having a D4Z4 repeat of <20 units on a 4qA allele results in transcriptional derepression of *DUX4*[21,23,38]. To further study the mechanism of repression imposed by SMCHD1 and LRIF1 at D4Z4 in somatic cells, we employed CRISPR/Cas9 genome editing to generate somatic knockout conditions for SMCHD1 (SMCHD1[KO]), the long isoform of LRIF1 (LRIF1L[KO]) or both isoforms of LRIF1 (long + short isoform, hereafter referred to as LRIF1L + S[KO]) in control immortalized myoblast cell lines derived from two unrelated control individuals (Fig. 1a, Supplementary Fig. 1a). Multiple studies have previously pointed out a relationship between D4Z4 repeat length, the degree of repeat hypomethylation and clinical severity, suggesting a role of a repeat length itself as a disease modifier[26,27,48,49]. Therefore, we chose one cell line with a 32-unit long 4qA FSHD permissive allele (control[32U]; beyond the typical FSHD2 repeat size range) and one with a 13-unit long 4qA allele (control[13U]; within the typical FSHD2 repeat size range). Interestingly, upon myogenic differentiation of the knockout clones, only the SMCHD1[KO] condition leads to significant *DUX4* transcriptional derepression in both control cell lines (Fig. 1b). However, the level of derepression was only mild as compared to *DUX4* expression measured in myogenic cell lines originating from two different FSHD2 individuals with heterozygous *SMCHD1* mutations (Fig. 1b). Furthermore, the expression of selected DUX4 target genes did not increase, suggesting that under these conditions, the DUX4 levels were insufficient to cause a significant transcriptional response of its target genes. *DUX4* expression is positively influenced by myogenic differentiation[38]. *MYH3* mRNA levels and fusion index, markers of myogenic differentiation, did not reveal major differences between WT and KO clones (Supplementary Fig. 1b, c) except in the case of SMCHD1[KO], which showed a mild significant increase in fusion index. This rules out the possibility that knockout of SMCHD1 or LRIF1 profoundly affects myogenesis, which would confound a direct effect on *DUX4* expression.

**Somatic loss of LRIF1 or SMCHD1 in control myoblasts does not result in D4Z4 chromatin changes typical for FSHD2.** The lack of a robust transcriptional *DUX4* response upon *SMCHD1* or *LRIF1* knockout prompted us to investigate the D4Z4 chromatin features characteristic of FSHD2 D4Z4 alleles. First, we examined DNA methylation as germline defects in SMCHD1 or LRIF1 in FSHD2 lead to pronounced pan-D4Z4 hypomethylation (combined 4q and 10q D4Z4 repeats), especially of 19 CpGs within the previously reported DR1 region and of 11 CpGs in the FAS-PAS region[23,49,50]. While the DR1 region is located proximal to the *DUX4* promoter in every repeat unit of chromosomes 4 and 10, the FAS-PAS region is specific for the distal part of 4qA alleles (Supplementary Fig. 2a). We analyzed three independent clones from each control[32U] and control[13U] knockout condition. Bisulfite PCR of the DR1 and FAS-PAS region followed by subcloning and sequencing did not, however, reveal noticeable changes in

either overall DNA methylation levels (Fig. 2a, b) or at individual CpGs in the DR1 or FAS-PAS amplicon in any of the knockout conditions compared to the WT situation (Supplementary Fig. 2b, c). This finding corroborates and extends on a previous study showing that SMCHD1 knockout in HEK293T embryonic kidney or HCT116 colon carcinoma cells does not result in D4Z4 hypomethylation[41].

Next, we performed chromatin immunoprecipitation of histone H3 and three of its histone marks (H3K9me3, H3K4me2 and H3K27me3) at three established regions within the 4q and 10q D4Z4 units, which are known to be deregulated at D4Z4 in FSHD2 due to germline mutations in *SMCHD1* or *LRIF1*[23,38,45]. As was the case for DNA methylation, *SMCHD1* and *LRIF1* somatic knockouts in control[32U] did not show altered levels of histone H3 itself (Fig. 2c) or any of the examined H3-associated histone modifications as compared to WT clones (Fig. 2d). Similar observations, i.e. largely unchanged H3K9me3 levels at D4Z4, have been made upon *SMCHD1* knockout in HCT116 cells[41]. These findings may thus explain the observed limited transcriptional response of the *DUX4* locus resulting from the absence of either SMCHD1 or LRIF1 in control myocytes since the examined repressive mechanisms in the form of DNA methylation and repressive histone modifications remained intact.

**LRIF1 recruitment to D4Z4 in somatic cells is partially SMCHD1-dependent, while SMCHD1 recruitment to D4Z4 is independent of LRIF1.** Independent proteomic studies aimed at identifying factors associated with specific histone modifications revealed an association of LRIF1 and SMCHD1 with H3K9me3[51–53]. At D4Z4, it was shown that reducing H3K9me3 levels in control myoblasts results in reduced SMCHD1 occupancy, placing SMCHD1 downstream of H3K9me3[54]. In mouse embryonic stem cells, a predominant mechanism for Smchd1 recruitment to H3K9me3-marked chromatin depends on Lrif1 and the same study proposed that LRIF1 could also mediate SMCHD1 recruitment to D4Z4 as LRIF1 recognizes HP1-bound H3K9me3 enriched heterochromatin[30]. To test if this proposed LRIF1-dependent SMCHD1 chromatin recruitment to D4Z4 mechanism holds, we performed SMCHD1 and LRIF1 chromatin immunoprecipitation in our somatic SMCHD1 and LRIF1 knockout control[32U] clones, where the H3K9me3 levels at D4Z4 are preserved (Fig. 2d). This allowed us to interrogate the interdependency of these two factors in their D4Z4 recruitment and the H3K9me3-dependency of this mechanism. In agreement with a previous study[41], SMCHD1 is mostly enriched at the DR1 region of D4Z4 with a gradual decrease in 3' direction (Q and Hox regions) in the WT situation and this enrichment pattern is also observed for LRIF1 (Fig. 3a). Interestingly, we did not observe reduced SMCHD1 binding to D4Z4 in either LRIF1 knockout condition (Fig. 3a). Therefore, the presence of SMCHD1 at D4Z4 in control[32U] cells with unperturbed D4Z4 heterochromatin is independent of LRIF1. On the other hand, we detected decreased LRIF1 enrichment at D4Z4 in SMCHD1[KO] cells to the same degree as in LRIF1L + S[KO] cells, which served as a baseline for the ChIP antibody background, suggesting that the presence of LRIF1 at D4Z4 is at least in part SMCHD1-dependent. Since H3K9me3 and DNA methylation levels at D4Z4 were not reduced in SMCHD1[KO] cells (Fig. 2d), this implies that H3K9me3 and DNA methylation are not sufficient for LRIF1 recruitment to D4Z4.

To study if the interdependency of SMCHD1 and LRIF1 D4Z4 chromatin binding has a role in FSHD2 pathogenesis, we performed SMCHD1 and LRIF1 ChIP-qPCR experiments in control and SMCHD1-mutant FSHD2 human primary myoblasts

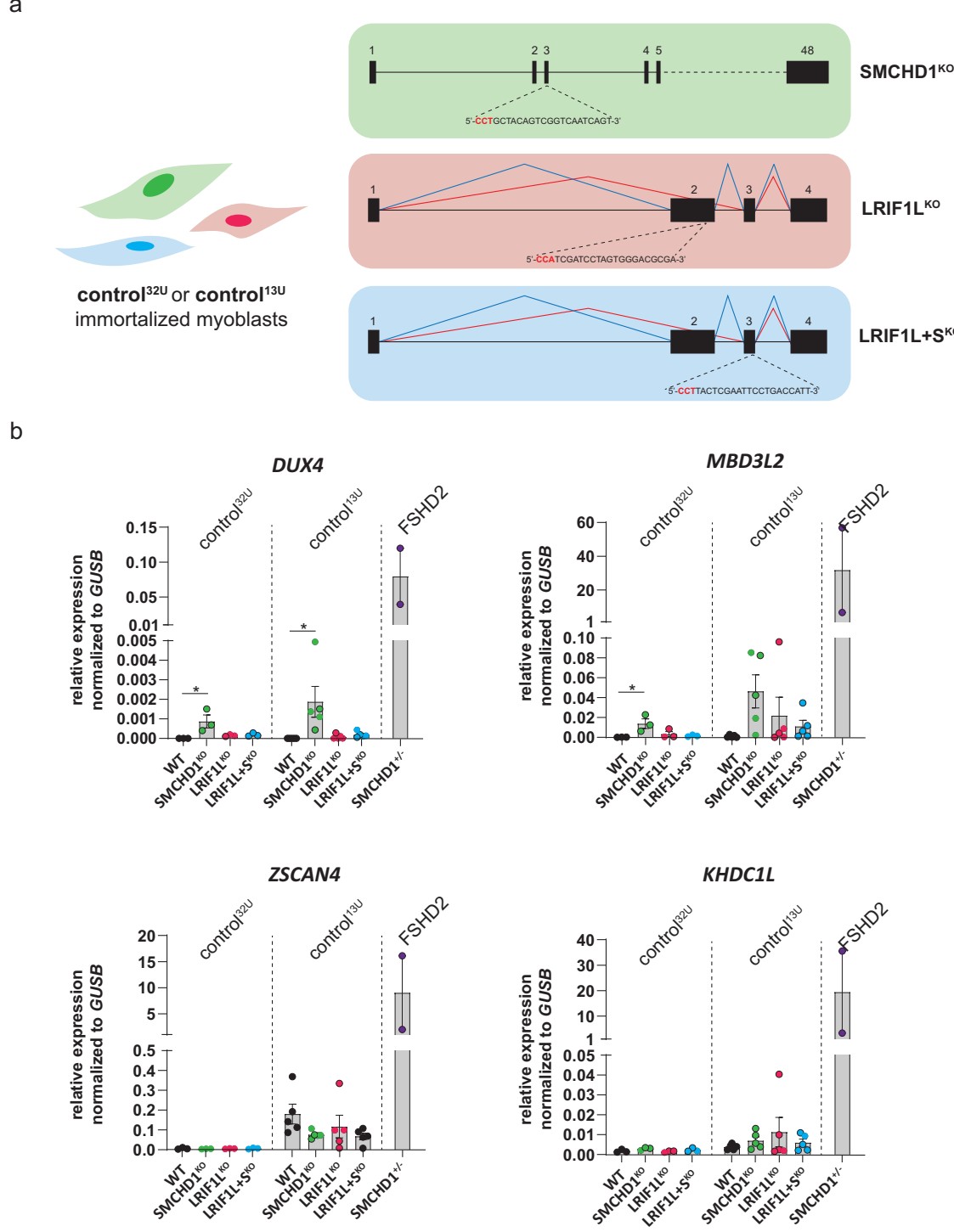

**Fig. 1 Knock-out of *SMCHD1* or *LRIF1* in control immortalized myoblasts have only a mild effect on *DUX4* derepression. a** Gene structure of human *SMCHD1* (top) and *LRIF1* (bottom) and the position of the sgRNAs used for creating respective KOs (PAM sequence labeled in red). Two different LRIF1 isoforms are produced by differential splicing of exon 2 as denoted by different splicing patterns (blue = long isoform, red = short isoform). **b** RT-qPCR of *DUX4*, three of its target genes (*MBD3L2*, *KHDC1L* and *ZSCAN4*) in differentiated WT and knockout clones derived from immortalized control[32U] and control[13U] myogenic lines. Bars represent mean ± SEM. Each dot represents one clone (three independent clones per genotype in control[32U] line and five independent clones per genotype in control[13U] line). For comparison, RT-qPCR was also performed on two independent FSHD2 immortalized myogenic lines. Statistical significance between WT and KO groups was calculated by one-way ANOVA with Dunnett's post hoc test (**$p < 0.01$, *$p < 0.05$, ns not significant).

(Supplementary Fig. 3). As expected, we detected a significantly decreased abundance of SMCHD1 in FSHD2 samples compared to controls at the studied D4Z4 regions. LRIF1 recruitment did not decrease significantly but showed a robust reduction in three

of the four studied FSHD2 samples suggesting that LRIF1 recruitment depends on SMCHD1 in FSHD2 patient samples.

To further examine the SMCHD1-dependency of LRIF1 at D4Z4, we studied the reverse situation and tested whether

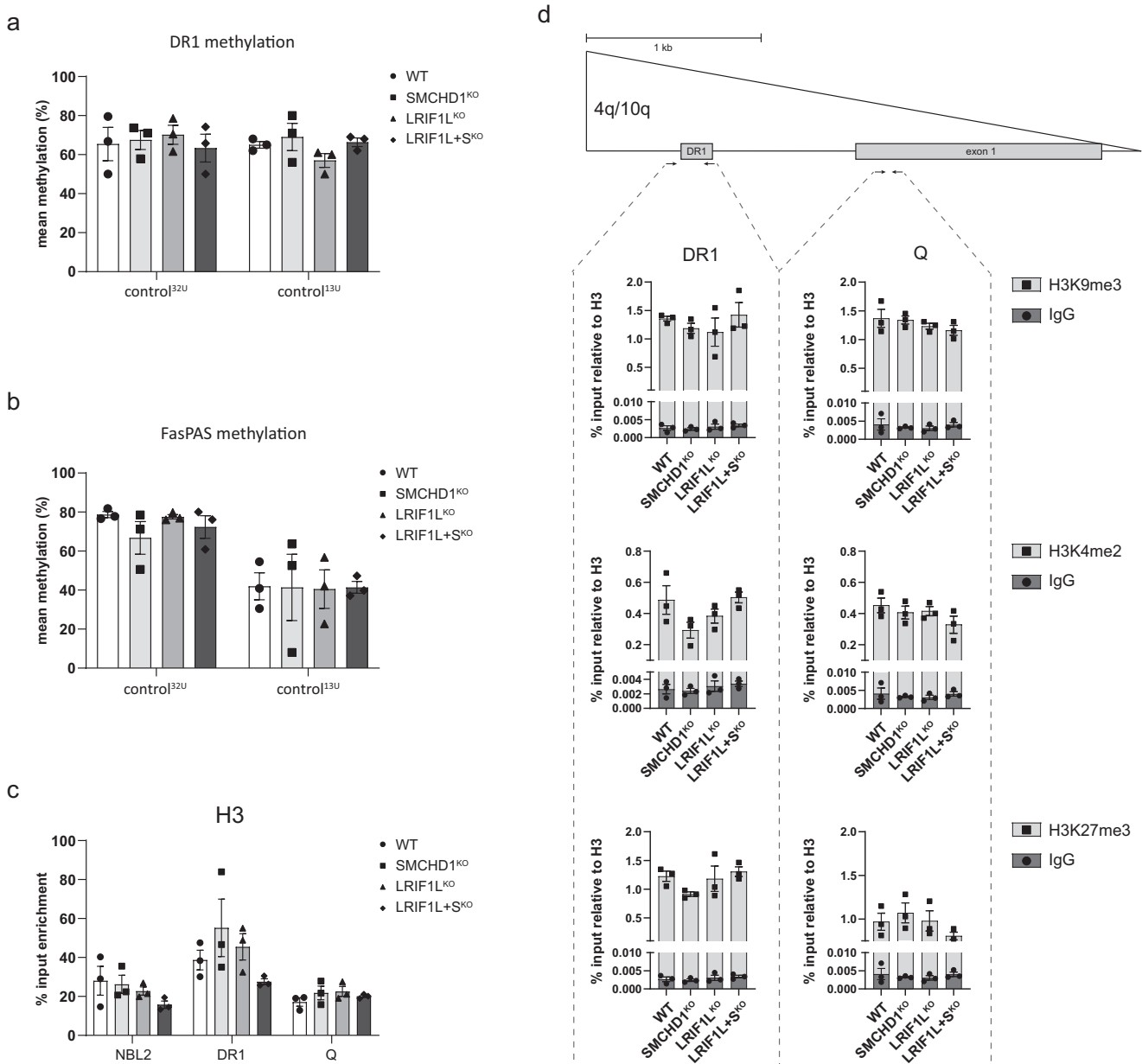

**Fig. 2 Somatic SMCHD1 and LRIF1 knockouts do not recapitulate perturbations of heterochromatin marks known in FSHD2. a** Mean DNA methylation level of the D4Z4 DR1 region (on 4q + 10q D4Z4 alleles) in different knockout clones derived either from control[32U] or control[13U] as determined by bisulfite Sanger sequencing. Bars and whiskers represent mean ± SEM of three independent clones per each genotype, respectively. **b** Mean DNA methylation level of the D4Z4 FasPAS region (4qA-specific) in different knockout clones derived from control[32U] or control[13U] as determined by bisulfite Sanger sequencing. Bars and whiskers represent mean ± SEM of three independent clones per genotype. **c** ChIP-qPCR of histone H3 at the DR1 and Q region in different control[32U] knockout conditions. Isotype specific IgG served as a background control. **d** ChIP-qPCR of selected H3 modifications at the D4Z4 DR1 and Q region in different control[32U] knockout conditions. Isotype specific IgG served as a background control. Schematic of a D4Z4 unit with position of the DR1 and Q region within D4Z4 examined by ChIP-qPCR indicated. Bars represent mean ± SEM (ns = 3 per genotype). Statistical significance between WT and KO groups was calculated by one-way ANOVA with Dunnett's post hoc test (ns not significant).

increased SMCHD1 binding to D4Z4 results in increased LRIF1 binding. We used a previously described FSHD2 muscle cell line, which carries a heterozygous germline mutation (c.4347-236 A > G) in intron 34 of the *SMCHD1* locus[47]. This mutation creates a cryptic splice site which leads to exonization of 53 bp of intronic sequence thereby disturbing the open reading frame of SMCHD1 and causing its haploinsufficiency (Fig. 3b). We recently showed that this genetic lesion can be corrected in myocytes by removing the pseudo-exon with a dual Cas9 strategy,

which restores SMCHD1 splicing and protein levels and results in *DUX4* suppression[47]. We performed chromatin immunoprecipitation studies of SMCHD1 and LRIF1 in four *SMCHD1* uncorrected and four corrected clones, which were previously characterized[47]. We found increased SMCHD1 enrichment at D4Z4 in the corrected myocyte clones, with the strongest rescue being at the DR1 region, thus explaining the previously observed *DUX4* repression in the corrected cells (Fig. 3c). Next, we tested whether this increased SMCHD1 binding was associated with

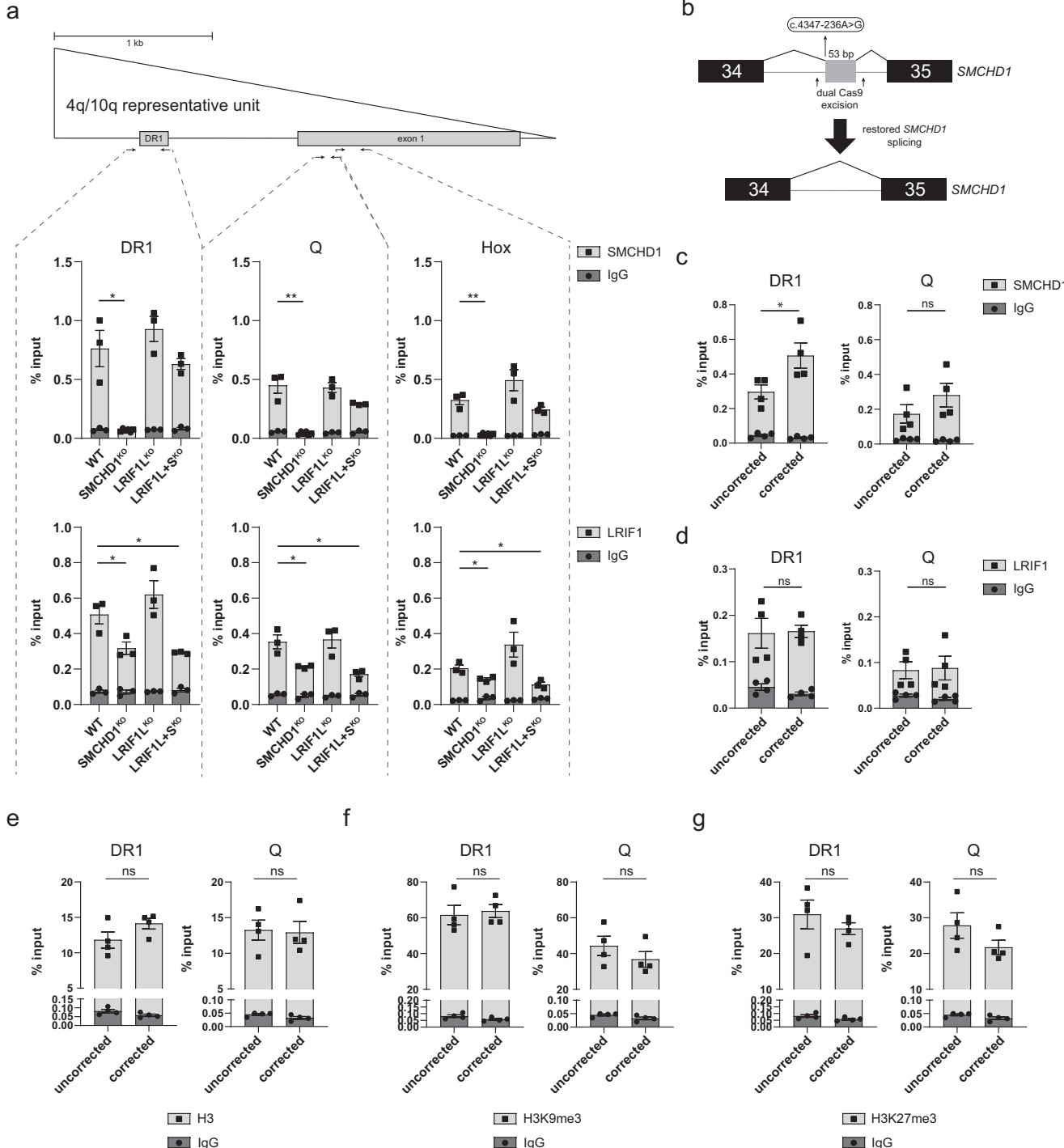

**Fig. 3 SMCHD1 binding to D4Z4 in somatic cells is independent of LRIF1. a** SMCHD1 and LRIF1 ChIP-qPCR in different control[32U] knock-out conditions. Schematic of one D4Z4 unit and the position of three regions within D4Z4 examined by ChIP-qPCR is indicated. Bars and whiskers represent mean ± SEM of three independent clones. Isotype specific IgG was used for background control. Statistical significance between WT and KO groups was calculated by one-way ANOVA with Dunnett's post hoc test (**$p < 0.01$, *$p < 0.05$). Only significant $p$-values are shown. **b** Schematic representation of splicing of the mutant *SMCHD1* allele carrying the intronic SNP variant indicated in the box. **c** SMCHD1 ChIP-qPCR of two D4Z4 regions (DR1 and Q) from four *SMCHD1* intron unedited and four *SMCHD1* intron edited clones that restores WT *SMCHD1* splicing. Bars and whiskers represent mean ± SEM. **d** LRIF1 ChIP-qPCR of two D4Z4 regions (DR1 and Q) from four *SMCHD1* intron unedited and four *SMCHD1* intron edited clones. Bars and whiskers represent mean ± SEM. Statistical significance was calculated with unpaired $t$-test (**$p < 0.01$, *$p < 0.05$, ns not significant). **e** H3 ChIP-qPCR of the D4Z4 DR1 and Q region from four *SMCHD1* unedited and four *SMCHD1* intron edited clones. Bars and whiskers represent mean ± SEM. Isotype specific IgG was used for background control. **f** H3K9me3 ChIP-qPCR of the D4Z4 DR1 and Q region from four *SMCHD1* unedited and four *SMCHD1* intron edited clones. Bars and whiskers represent mean ± SEM. Isotype specific IgG was used for background control. **g** H3K27me3 ChIP-qPCR of the D4Z4 DR1 region from four *SMCHD1* unedited and four *SMCHD1* intron edited clones. Bars and whiskers represent mean ± SEM. Isotype specific IgG was used for background control. Statistical significance was calculated with an unpaired $t$-test (ns not significant).

increased LRIF1 binding to D4Z4. However, LRIF1 enrichment at D4Z4 at two examined sites (DR1 and Q) did not change significantly in *SMCHD1* corrected clones (Fig. 3d).

To investigate why increased SMCHD1 levels did not restore LRIF1 enrichment at D4Z4, we examined the chromatin state of D4Z4 in *SMCHD1* corrected clones. We previously showed that restoring SMCHD1 levels in these corrected clones does not lead to re-methylation of D4Z4[47]. Further examination of the H3K9me3 and H3K27me3 histone modifications showed that correction of the *SMCHD1* mutation did not result in the re-establishment of a D4Z4 histone modification pattern observed in healthy individuals (Fig. 3e, f, g). These results suggest that modulating SMCHD1 levels in somatic cells represses *DUX4* without affecting known features of D4Z4 chromatin: DNA methylation, H3K9 trimethylation, H3K27 trimethylation and LRIF1 binding. Indeed, the data suggest that LRIF1 binding to D4Z4 does not solely depend on SMCHD1 and that other chromatin factors or marks play a role in somatic cells.

**SMCHD1 and LRIF1 provide auxiliary repression of *DUX4* at epigenetically compromised D4Z4 repeats.** Since modulating SMCHD1 levels in FSHD2 myoblasts affects *DUX4* levels, we further explored this in two unrelated conditions in which the D4Z4 repeat is hypomethylated due to either dysfunctional DNA methylation maintenance or its establishment. This allowed us to assess if SMCHD1 and LRIF1 can bind to hypomethylated D4Z4 repeats and enforce *DUX4* repression in a situation where the epigenetic disturbance of D4Z4 is not due to germline mutations in *SMCHD1* or *LRIF1*. First, we focused on a situation in which hypomethylated D4Z4 arose due to inactivation of the DNA methylation maintenance machinery. We used the colorectal cancer line HCT116 and its *DNMT1* and *DNMT3B* double knockout (DKO) derivative[55]. D4Z4 hypomethylation in HCT116 DKO cells is accompanied by a reduction in H3K9me3 and gain in H3K4me2, ultimately resulting in *DUX4* derepression since it also has a permissive 4qA allele[16]. Somatic loss of DNA methylation in HCT116 DKO cells leads to 5' to 3' redistribution of SMCHD1 along the D4Z4 unit (Supplementary Fig. 4), in agreement with previous findings[41]. Similarly, the LRIF1 enrichment pattern followed the one of SMCHD1 (Supplementary Fig. 4).

Next, we used a cellular model system in which the D4Z4 repeat is hypomethylated in somatic cells derived from individuals with germline mutations in *DNMT3B*, thus representing a case of compromised DNA methylation establishment at D4Z4. For this we studied primary fibroblasts from individuals having either heterozygous (*DNMT3B*^het) or biallelic *DNMT3B* mutations (*DNMT3B*^bi). All *DNMT3B*^bi fibroblasts are derived from individuals diagnosed with ICF1 syndrome. These individuals present with more pronounced D4Z4 hypomethylation compared to their heterozygous unaffected relatives[22]. First, we characterized the D4Z4 chromatin in these samples to examine if the DNA hypomethylation is accompanied by histone modification changes typical for FSHD2. We performed ChIP-qPCR for H3K4me2, H3K9me3 and H3K27me3 and histone H3. Already H3 itself was reduced compared to primary fibroblasts from control individuals suggesting a possible loosening or remodeling of nucleosomes at D4Z4 in fibroblasts from individuals with mono- or biallelic *DNMT3B* mutations (Supplementary Fig. 5a). In addition, H3K9me3 levels were decreased while those of H3K4me2 and H3K27me3 were increased, similar to what has been observed in FSHD2 fibroblasts carrying either *SMCHD1* or *LRIF1* mutations (Supplementary Fig. 5b). Interestingly, DNMT3B^het fibroblasts displayed an intermediate phenotype between WT and DNMT3B^bi fibroblasts. Next, we performed

ChIP-qPCRs for SMCHD1 and LRIF1. We also included primary FSHD2 fibroblasts, which have heterozygous SMCHD1 mutations (SMCHD1^het), and in which SMCHD1 and LRIF1 occupancy at D4Z4 is expected to be reduced based on the previous studies[21,23,38]. Interestingly, whereas the SMCHD1 and LRIF1 D4Z4 enrichment profile in HCT116 DKO cells showed evidence for a redistribution (Supplementary Fig. 3), in primary DNMT3B mutant fibroblasts their occupancy was reduced at all three tested D4Z4 regions with the strongest impact observed at the D4Z4 DR1 site, while at the Q and Hox region the enrichment difference did not reach statistical significance (Fig. 4a). SMCHD1 and LRIF1 enrichment levels in DNMT3B^het and DNMT3B^bi fibroblasts were similar to enrichment levels measured in SMCHD1^het (FSHD2) fibroblasts. Altogether, this shows that both SMCHD1 and LRIF1 recruitment to D4Z4 is sensitive to chromatin changes associated with DNA hypomethylation either at the somatic stage as represented by the results from our studies in HCT116 DKO cells or by a failure in DNA methylation establishment during early development as represented by the results from our studies in fibroblasts carrying *DNMT3B* mutations.

Additionally, we tested if there is a synergistic effect of heterochromatin marks and SMCHD1 and LRIF1 on *DUX4* repression. We used a mix of siRNAs to deplete SMCHD1, LRIF1L or LRIF1L + S in ICF1 proliferating myoblasts (Rf285.3) derived from an individual who carries an 11 D4Z4 unit-long permissive 4qA allele (Fig. 4b). Since these myoblasts have biallelic mutations in the *DNMT3B* gene, the D4Z4 heterochromatin is already compromised, as shown above. All three knock-down scenarios lead to variable degrees of *DUX4* transcriptional upregulation and activation of four *DUX4* target genes (*ZSCAN4*, *KHDC1L*, *TRIM43*, *MBD3L2*) as compared to cells treated with a non-targeting siRNA mix (Fig. 4c). This suggests that despite the decreased SMCHD1 and LRIF1 enrichment at D4Z4 in ICF1, these proteins still provide residual repression, and their depletion leads to further *DUX4* transcriptional derepression.

**SMCHD1 and the long isoform of LRIF1 negatively regulate *LRIF1* expression.** Lastly, to evaluate a genome-wide repressive function of SMCHD1 and LRIF1, we performed poly-A RNA-seq in WT and respective knockout clones derived from the control^32U line. For this, we used undifferentiated myocytes to avoid transcriptional differences which could arise due to different myogenic differentiation of individual clones, as well as to prevent any possible DUX4-driven signature as *DUX4* is expressed, albeit at low levels, in myotubes of the knockout clones (Fig. 1c). Differential expression analysis did not reveal major transcriptional changes in any of the knockout conditions when compared to WT clones, with SMCHD1^KO having the strongest impact out of the three knockout conditions (Fig. 5a, Supplementary Fig. 6a, b, Supplementary Data 1). These results extend on the previously reported lack of transcriptional deregulation after siRNA-mediated knock-down of SMCHD1 or LRIF1L + S in female RPE1-hTERT cells[46] and suggest that neither short-term, nor permanent depletion of SMCHD1 or LRIF1 in somatic cells has a major impact on the transcriptome.

Interestingly, we noticed that the *LRIF1* gene was upregulated in SMCHD1^KO clones. Furthermore, it was not downregulated in either of the two LRIF1^KO situations as might be expected from the CRISPR/Cas9 induced indels leading to a premature stop codon which triggers non-sense mediated decay (NMD) of transcripts. This is for example the case for *SMCHD1* transcripts in the SMCHD1^KO condition (Fig. 5a). We validated this observation by RT-qPCR using exon junction primers specifically detecting *LRIF1* long (ex2-3), short (ex1-ex3) or all isoforms (ex3-4) (Fig. 5b). Indeed, we detected

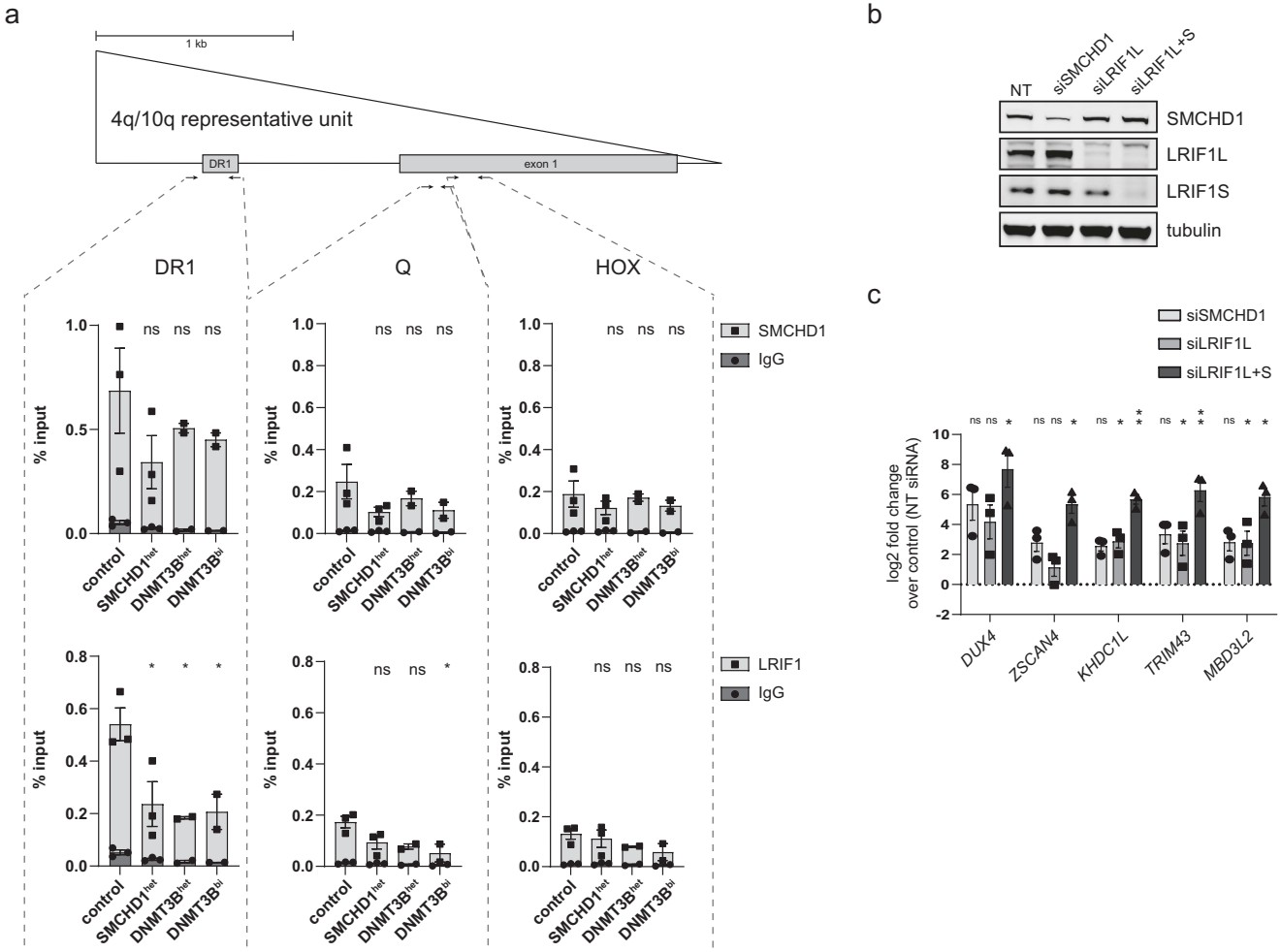

**Fig. 4 SMCHD1 and LRIF1 have residual repressive action at hypomethylated D4Z4. a** SMCHD1 and LRIF1 ChIP-qPCR in primary control ($n = 3$, lines: 2524, 2397, 2333) fibroblasts and fibroblasts carrying either a heterozygous *SMCHD1* mutation ($n = 3$, lines: 2440, 2337, 2332), a heterozygous *DNMT3B* mutation ($n = 2$, lines: v294, b974) or biallelic *DNMT3B* mutations ($n = 2$, lines: Rf699.3, Rf286.3). Schematic of one D4Z4 unit in which the position of three regions within D4Z4 examined by ChIP-qPCR is indicated (DR1, Q, HOX). Bars and whiskers represent mean ± SEM. Isotype specific IgG was used for background control. Statistical significance between WT and mutant groups was calculated by one-way ANOVA with Dunnett's post hoc test (*$p < 0.05$, ns not significant). **b** Western blot confirmation of successful siRNA-mediated knock-down of SMCHD1, LRIF1L or LRIF1L + S in primary ICF1 myoblasts. Tubulin was used as a loading control. **c** RT-qPCR of *DUX4* and four of its target genes (*ZSCAN4, KHDC1L, TRIM43* and *MBD3L2*) after siRNA-mediated KD of SMCHD1, LRIF1L or LRIF1L + S in ICF1 myoblasts. Expression levels detected in KD cells were normalized to cells transfected with non-targeting (NT) siRNA and further log2 transformed. *GUSB* was used as a housekeeping gene for intra-sample normalization. Bars and whiskers represent mean ± SEM of three independent experiments. Statistical significance was calculated by one sample *t*-test (**$p < 0.01$, *$p < 0.05$, ns not significant).

elevated transcript levels of both *LRIF1* isoforms in SMCHD1[KO] clones and even increased levels of the *LRIF1* short isoform in LRIF1L[KO] clones. This prompted us to examine if LRIF1 and SMCHD1 directly regulate the *LRIF1* locus. Examining previously published SMCHD1 and LRIF1 ChIP-seq datasets from hTERT-immortalized retinal pigment epithelial (RPE1) cells[46] revealed enrichment of both SMCHD1 and LRIF1 immediately upstream of *LRIF1* exon 1, coinciding with the CpG island (Fig. 5c). We confirmed this ChIP-seq peak with SMCHD1 and LRIF1 ChIP-qPCR also in control[32U] cells suggesting that this transcriptional regulation might be conserved between different cell types (Fig. 5d, e). The enrichment of both proteins is reduced in SMCHD1[KO] clones (Fig. 5d, e) and already in LRIF1L[KO] cells, which still express the short isoform (Fig. 5e). Furthermore, there was no further reduction in SMCHD1 or LRIF1 enrichment LRIF1L + S[KO] cells suggesting that there is no synergistic effect of the two LRIF1 isoforms at this locus. This differs from the situation at D4Z4 where SMCHD1 binding is affected neither in LRIF1L[KO] nor in LRIF1L + S[KO] clones

(Fig. 3a). In addition, the overall enrichment of LRIF1 is not affected at D4Z4 in LRIF1L[KO] cells in contrast to the situation at the *LRIF1* promoter. This suggests the existence of different binding properties of LRIF1 isoforms to these two loci.

Promoters of expressed genes are known to be decorated by active histone marks such as H3K4me3 and H3K4me2, while promoters of silent genes are marked with repressive histone marks such as H3K9me3 and H3K27me3. As LRIF1 and SMCHD1 are known to be associated with H3K9me3 and we show that both proteins bind to the LRIF1 promoter in control[32U] cells, we examined the histone marks at this locus. Interestingly, despite the *LRIF1* gene being expressed in these cells, its promoter is characterized by the active H3K4me2 mark and by the repressive histone marks H3K9me3 and H3K27me3, as opposed to the promoter of the constitutively expressed housekeeping gene *GAPDH*, which is only enriched for the active H3K4me2 mark (Supplementary Fig. 6c). As *LRIF1* expression is upregulated in SMCHD1 and LRIF1 knockout control[32U] cells,

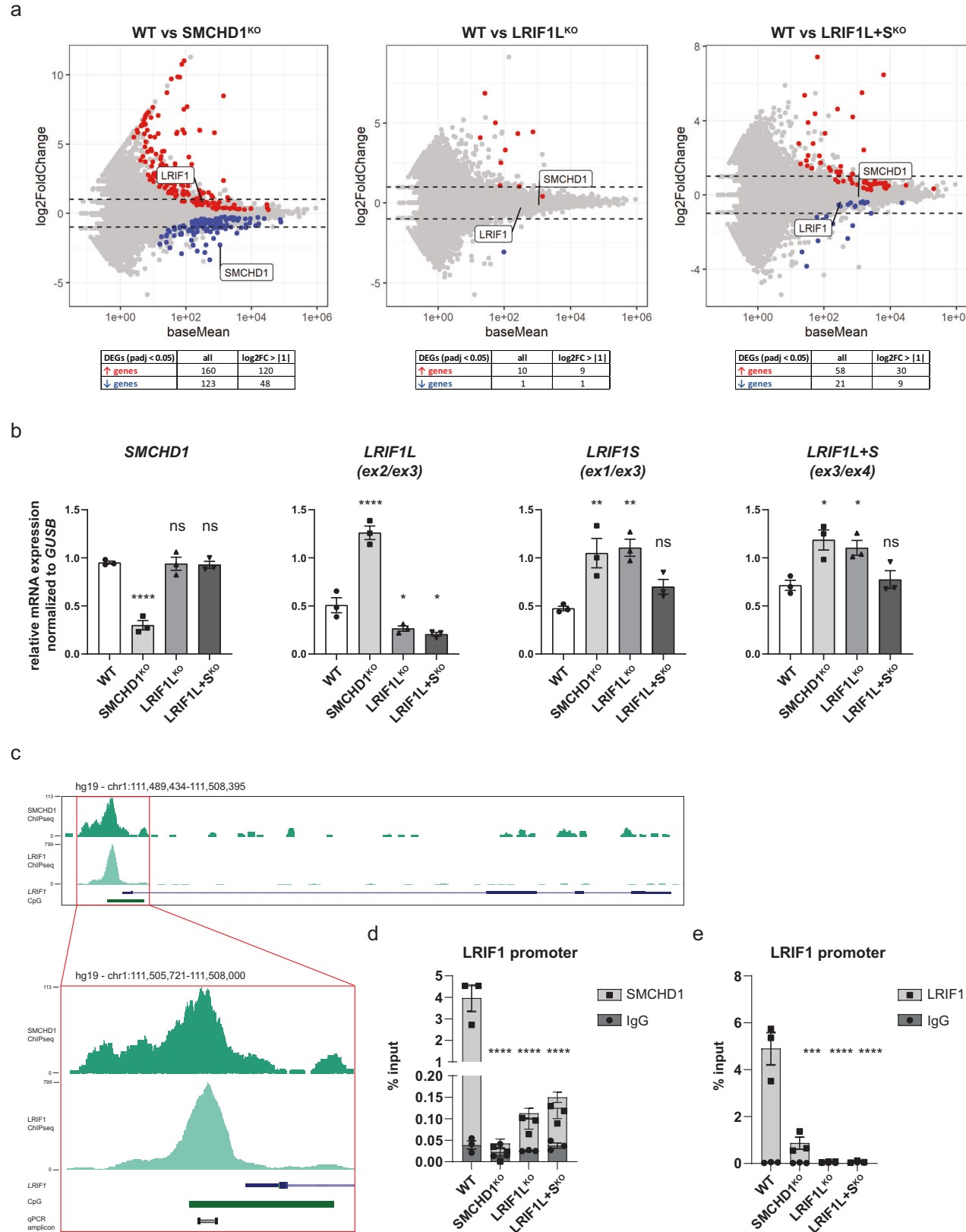

we wondered if we can find underlying changes in histone marks which would explain such transcriptional response, possibly increased levels in active marks and/or decreased levels of repressive marks. Surprisingly, the H3 level and all examined histone marks coupled to this histone was reduced in each knockout condition (Supplementary Fig. 6d). This indicates that

the observed transcriptional upregulation could be due to a nucleosome displacement in the *LRIF1* promoter.

Lastly, since somatic depletion of SMCHD1 in control[32U] cells results in reduced LRIF1 binding at the *LRIF1* promoter and subsequent *LRIF1* upregulation, we examined if SMCHD1 and LRIF1 enrichment at the *LRIF1* promoter would also be

**Fig. 5 SMCHD1 and LRIF1 long isoform negatively regulate *LRIF1* expression. a** MA plots of RNA-seq experiments performed on three independent WT, SMCHD1[KO], LRIF1L[KO] or LRIF1L + S[KO] clones derived from the control[32U] muscle cell line. Differentially upregulated genes are highlighted in red and differentially downregulated genes are in blue (*p*-adjusted < 0.05) with summary of differentially expressed genes provided in a table format below each MA plot. Dashed lines mark log$_2$ fold change of |1|. *SMCHD1* and *LRIF1* transcripts are indicated. **b** RT-qPCR of *SMCHD1* and different exon junctions of *LRIF1* to differentiate between expression of different *LRIF1* isoforms (ex2-ex3 = long isoform, ex1-ex3 = short isoform, ex3-4 = both isoforms). Bars and whiskers represent mean ± SEM. Each dot represents one clone. Statistical significance between WT and KO groups was calculated by one-way ANOVA with Dunnett's post hoc test (****<0.0001, ***<0.001, **$p$ < 0.01, *$p$ < 0.05, ns not significant). **c** SMCHD1 and LRIF1 ChIP-seq from RPE1 cells showing SMCHD1 and LRIF1 enrichment over the LRIF1 promoter region. A zoom of the promoter region is presented to depict the amplicon used for ChIP-qPCR. **d** SMCHD1 ChIP-qPCR of the *LRIF1* promoter in different control[32U] WT and KO conditions. Bars and whiskers represent mean ± SEM of three independent clones. Isotype specific IgG was used as background control. **e** LRIF1 ChIP-qPCR of the *LRIF1* promoter in different control[32U] WT and KO conditions. Bars and whiskers represent mean ± SEM of three independent clones. Isotype specific IgG was used as background control. Statistical significance between WT and KO groups was calculated by one-way ANOVA with Dunnett's post hoc test (****<0.0001, ***<0.001).

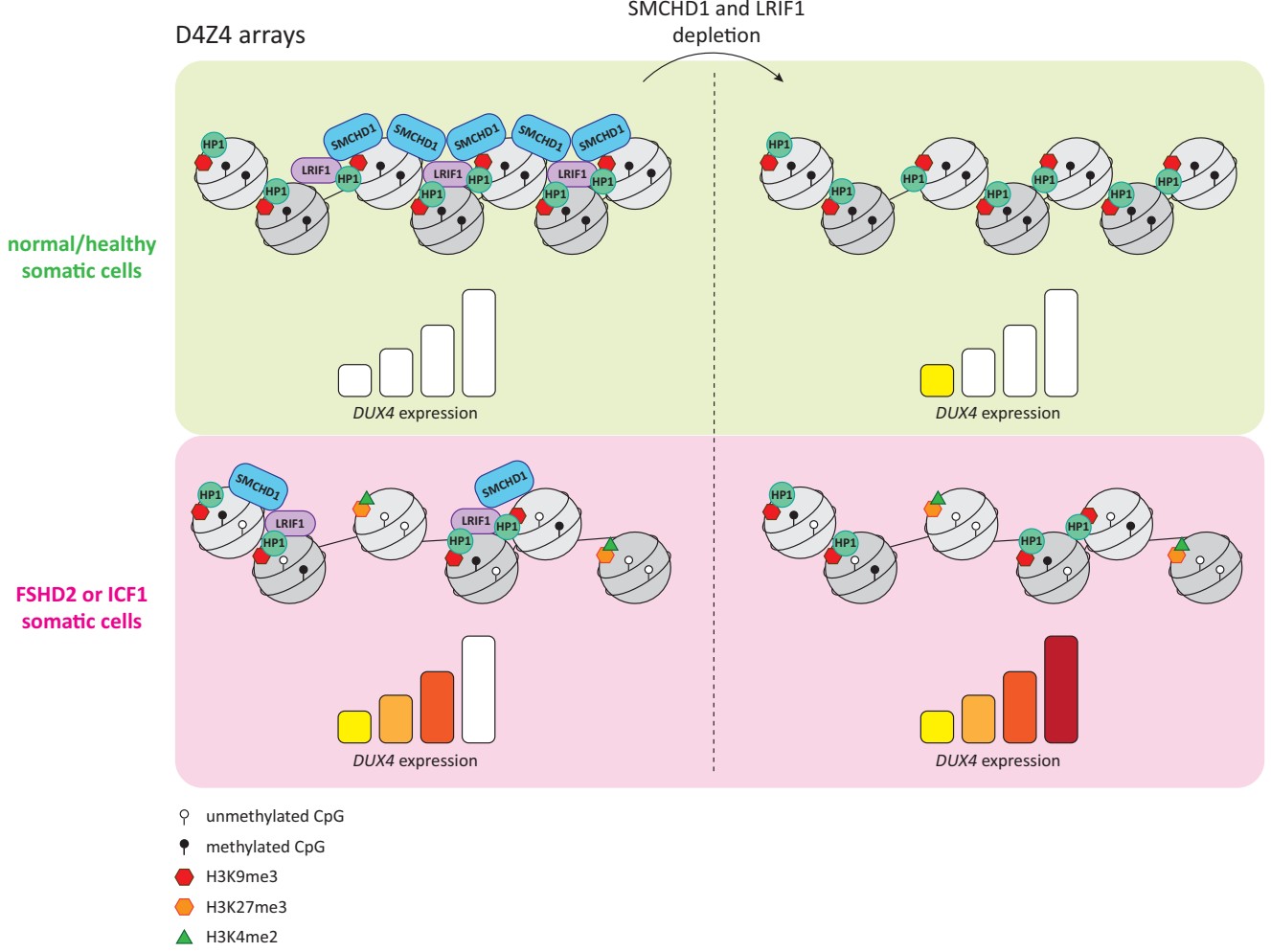

**Fig. 6 Model for SMCHD1 and LRIF1-mediated D4Z4 repression in somatic cells.** In somatic cells from unaffected individuals, D4Z4 is marked by high levels of CpG methylation, H3K9me3, HP1 proteins as well as LRIF1 and SMCHD1, resulting in transcriptional silencing of *DUX4*. LRIF1 recruitment is partially dependent on SMCHD1 and possibly stabilized by HP1, while the mechanism of SMCHD1 recruitment to D4Z4 requires H3K9me3. In ICF1 or FSHD2 somatic cells, both SMCHD1 and LRIF1 occupancy is reduced at D4Z4 due to lower H3K9me3 resulting in *DUX4* transcriptional derepression. Additional depletion of SMCHD1 or LRIF1 in both situations results in further *DUX4* upregulation.

decreased in FSHD2 primary fibroblasts with heterozygous *SMCHD1* mutations or in primary fibroblasts carrying either heterozygous or biallelic *DNMT3B* mutations similarly to what we observed at the D4Z4 repeat. The ChIP-qPCR of the *LRIF1* promoter in the same set of primary fibroblasts as in Fig. 4a did not reveal differences in SMCHD1 or LRIF1 enrichment at this locus (Supplementary Fig. 6e, f). This observation is consistent with unchanged *LRIF1* expression in different examined cell types (primary fibroblasts, myoblasts or

differentiated myotubes) derived from FSHD2 individuals with SMCHD1 haploinsufficiency compared to their control counterparts (Supplementary Fig. 6g). These results suggest a different binding mechanism of SMCHD1 and LRIF1 to different genomic regions marked by H3K9me3 and different sensitivity of these regions to either germline or somatic dosages of these genes as evidenced by the expression regulation of the *DUX4* gene organized in a repetitive macrosatellite structure and the single-copy *LRIF1* locus.

## Discussion

To date, mutations in *SMCHD1*, *DNMT3B* and *LRIF1* have been identified to cause FSHD type 2, a disease in which the chromatin structure of the D4Z4 repeat is compromised, leading to inappropriate expression of *DUX4* in skeletal muscle. Therefore, understanding the role of these proteins in establishing or maintaining a repressive D4Z4 epigenetic landscape in somatic cells is not only important from a biological perspective, but also of clinical importance.

While the exact biological roles of SMCHD1 and LRIF1 are less defined, the function of DNMT3B is to establish the DNA methylation pattern during early embryonic development[56]. While the expression levels of the catalytically active isoform DNMT3B1 sharply decline during pluripotent stem cell differentiation, cells continue to express its catalytically inactive isoforms[57,58], albeit at low levels. These catalytically inactive isoforms are thought to act as accessory proteins to catalytically active DNMT1, thus aiding the DNA methylation maintenance process in somatic cells[59]. Interestingly, two studies reported a role for the catalytically active DNMT3B1 isoform in skeletal muscle cells[60,61]. However, whether catalytically active or inactive DNMT3B isoforms have a physiologically relevant function in D4Z4 repression in skeletal muscle cells after methylation patterns have been established in early embryogenesis, remains to be addressed. In contrast, we have previously demonstrated that SMCHD1 and LRIF1 have a *DUX4* expression modifying role in somatic cells, having observed that altering their levels in FSHD1 and FSHD2 myoblasts affects the expression of *DUX4* by yet unknown mechanisms[21,23,38]. This provided the rational to focus on the knockout of SMCHD1 and LRIF1 in somatic cells, preferentially in a myogenic context relevant to the disease, as epigenetic regulation is often highly tissue context-dependent.

Here, we aimed to study the role of SMCHD1 and LRIF1 in D4Z4 repression. To do so, we evaluated their repressive activity at D4Z4 in different genetic and chromatin contexts of D4Z4. First, we created SMCHD1 and LRIF1 knockout clones in two independent control immortalized myoblast lines with different D4Z4 repeat sizes and performed expression and chromatin studies of the D4Z4 repeat. Ablation of these factors from control cells which have undergone uncompromised epigenetic establishment trajectories during development, showed that these factors do not play a role in its heterochromatin maintenance once the D4Z4 epigenetic landscape is established although we cannot exclude that some cell adaptation to the chronic absence of SMCHD1 or LRIF1 has influenced our results. Rather, they provide an auxiliary molecular seal on top of the existing chromatin structure and increase the repression robustness of this locus against leaky transcription. This conclusion is supported by the observation that in *SMCHD1* gene-corrected FSHD2 patient myoblasts, cells in which *SMCHD1* haploinsufficiency is rescued, *DUX4* repression is achieved in the absence of a reversal of the chromatin landscape as determined by DNA methylation, H3K9me3, H3K27me3, and H3K4me2. These factors thus control *DUX4* expression by other yet unknown mechanism, possibly by promoting further chromatin condensation or higher-order chromatin conformation as was recently reported in mice for Smchd1 in the process of inactive X formation and *Hox* gene cluster regulation[31,34,35,62].

Earlier studies in mouse embryonic stem cells reported that Smchd1 recruitment to H3K9me3 enriched chromatin depends on Lrif1 and hypothesized a similar model of Smchd1 recruitment to H3K9me3 enriched D4Z4 in somatic cells[30]. In line with this hypothesis, we have previously observed decreased SMCHD1 levels at D4Z4 in somatic cells derived from an FSHD2 individual in whom the long isoform of LRIF1 is absent to similar levels as observed in FSHD2 cases with an *SMCHD1* defect[23]. In contrast,

here we show that knocking out specifically the long or both LRIF1 isoforms in control immortalized myoblasts does not affect SMCHD1 binding to D4Z4, which suggests that SMCHD1 recruitment, at least in somatic cells with properly established D4Z4 heterochromatin, is not dependent on LRIF1. On the other hand, we show that the loss of SMCHD1 in somatic cells leads to decreased LRIF1 enrichment at D4Z4 and similarly, that LRIF1 enrichment at D4Z4 is decreased in FSHD2 cases with *SMCHD1* mutations. However, LRIF1 recruitment to D4Z4 does not seem solely dependent on SMCHD1 as rescuing SMCHD1 levels in FSHD2 cells and increasing its levels at D4Z4 in its derepressed state does not lead to higher LRIF1 levels. This might suggest that LRIF1 recruitment to D4Z4 is dependent on some other chromatin factor apart from SMCHD1 that was not restored upon *SMCHD1* gene correction, such as H3K9me3 or factor(s) dependent on this mark like HP1 proteins.

D4Z4 is decorated with H3K9me3 in somatic cells and this repressive histone mark is significantly decreased at this locus in FSHD2, ICF1, and HCT116 DKO cells. Others have shown that the presence of this mark is crucial for SMCHD1 recruitment to D4Z4 in somatic cells[54] and thus could explain the decreased levels of SMCHD1 and LRIF1 at D4Z4 in its hypomethylated state as DNA hypomethylation concomitantly results in lower H3K9me3 levels as evidenced by results from HCT116 DKO and samples with heterozygous or homozygous *DNMT3B* mutations. The remaining H3K9me3 at hypomethylated D4Z4 could explain the residual SMCHD1 and LRIF1 binding to this locus and the previously observed reduced binding of SMCHD1 in cells from an FSHD2 individual in whom the LRIF1 long isoform is absent or from *DNMT3B* mutation carriers. In all these conditions the H3K9me3 mark is reduced at D4Z4. This also suggests a more fine-tuning role for SMCHD1 and LRIF1 in *DUX4* repression in somatic cells as correctly established D4Z4 repeat chromatin displays a large degree of resistance to its transcriptional derepression (Fig. 6). Similar observations regarding stability of DUX4 repression in healthy control cells have been made when treating cells with different epigenetic drugs which remove either DNA methylation or reduce H3K9me3 levels[48]. In contrast to FSHD1 myocytes, in which *DUX4* expression was further enhanced upon such pharmacological interventions, *DUX4* silencing in control cells remained intact or showed only minimal de-repression in case of one cell line derived from a healthy individual but which already showed leaky *DUX4* expression[48]. We have previously shown that *DUX4* expression was significantly upregulated also in control myogenic cell line when LRIF1L was knocked down with siRNAs[23], however, this cell line also already expressed low levels of *DUX4*, which could thus explain the discrepancy between previous results and current results obtained with knock-out lines which do not have any basal *DUX4* expression in their WT state.

We have also uncovered the regulation of the *LRIF1* locus by SMCHD1 and the long isoform of LRIF1 itself through their binding to the *LRIF1* promoter in somatic cells. This *LRIF1* gene regulation is likely not relevant for FSHD2 pathology since SMCHD1 nor LRIF1 binding was affected in FSHD2 cells which is consistent with the unchanged expression levels of *LRIF1* in these cells. This work thus extends our knowledge about the versatile involvement of SMCHD1 in regulating different types of chromatin (euchromatin as represented by *LRIF1* locus, facultative heterochromatin as represented by the inactive X chromosome[31,33,35,63–66] and tissue-specific expression attenuation of developmental genes such as clustered *PCDH*[65–68] or *HOX* genes[31,68], and constitutive heterochromatin exemplified by D4Z4). Interestingly, knocking out SMCHD1 in the control[32]U immortalized myoblasts cell line did not lead to dysregulated expression of clustered *PCDH* or *HOX* genes or genes on the

inactive X, which is consistent with findings obtained from near-diploid RPE1 cells upon SMCHD1 depletion[46] but opposed to findings from HEK293T cells, where SMCHD1 depletion lead to upregulation of *PCDH* β cluster and preferential upregulation of X chromosomal genes[69]. This emphasizes the importance of studying these proteins in disease-relevant tissues and begs the question what underlies this different sensitivity of SMCHD1-regulated loci to its gene dosage in early development versus in differentiated somatic stage as well as in different cell types.

## Methods

**Cell lines and culturing**. Primary and immortalized myoblasts were cultured in DMEM/F-10 medium (#31550, Gibco/Life Technologies) supplemented with 20% fetal bovine serum (FBS #10270, Gibco/Life Technologies), 1% Penicillin/Streptomycin (Pen/Strep #15140, Gibco/Life Technologies), with addition of 10 ng/ml rhFGF (#G5071, Promega) and 1 μM dexamethasone (#D2915, Sigma-Aldrich). Myoblasts were fused at 80% confluency by replacing growth medium with DMEM/F-12 Glutamax medium (#31331, Gibco/Life Technologies) containing 1% penicillin/streptomycin and 2% KnockOut serum replacement formulation (#10828, Gibco/Life Technologies) for 2–5 days depending on the cell line. The HEK293T cells were grown in Gibco DMEM, High Glucose, Pyruvate (#119950, Gibco/Life Technologies) with addition of 10% FBS and 1% Penicillin/streptomycin. Primary fibroblasts were cultured in DMEM/F-12 GlutaMAX™ Supplement (Gibco, #10565018) supplemented with 20% FBS, 1% penicillin/streptomycin, 10 mM HEPES (Gibco, #15630056) and 1 mM Sodium Pyruvate (Gibco, #11360070). The human colon carcinoma HCT116 (WT and DKO) cell lines were grown in McCoy's 5 A medium (Thermo Fisher Scientific, #16600082) supplemented with 10% FBS and 1% penicillin/streptomycin. Additional information about cell lines is provided in Supplementary Table 1.

**Generation of knockout cell lines with CRISPR/Cas9**. The sgRNA sequences targeting exon 3 of *SMCHD1*, exon 2 of *LRIF1* (LRIF1 long isoform specific knockout) or exon 3 of *LRIF1* (both LRIF1 isoforms knockout) were designed using the CRISPOR online design tool[70] (available at http://crispor.tefor.net/crispor.py). The sgRNA oligonucleotides (sequences in Supplementary Table 2) were cloned into the pX458 vector (Addgene #458138) via BbsI sites as described previously[71]. Immortalized myoblasts were seeded in 6-well plates to 60–70% confluency 1 day prior to transfection. Cells were transfected with 2.5 μg/well of pX458 vector containing gene-specific sgRNAs with Lipofectamine 3000 reagent according to the manufacturer instructions. 24 h after transfection medium was exchanged and 3 days post-transfection GFP positive cells were single-cell sorted to 96-well plates using a BD FACS Aria™ III cell sorter. Single cells were expanded and knockouts were confirmed by Western blot. As WT control clones were used single-cell sorted cells derived either from untransfected pool or a pool transfected with vector encoding only Cas9 without sgRNA.

**siRNA transfections**. One day prior transfection, $2 \times 10^5$ cells were seeded in 6-well plate. The next day, cells were transfected with 25 pmol of gene-specific siRNA mix using RNAiMAX (Thermo Fisher Scientific, #13778075) according to manufacturer's instructions. A non-targeting siRNA was used as a negative control. Cells were harvested 3 days post-transfection for respective analysis.

**RNA isolation, cDNA synthesis and RT-qPCR**. Cells were lysed in Qiazol (Qiagen, #79306) and total RNA was isolated with RNeasy mini kit (Qiagen, #74101) with on-column DNase I treatment. 1–2 μg of RNA was used for cDNA synthesis with poly-dT primer using RevertAid H Minus First Strand cDNA synthesis kit (Thermo Fisher Scientific, #K1621). Gene expression was analyzed in technical triplicates using iQ™ SYBR® Green Supermix (Biorad, #1708887) on CFX384 Touch Real-Time PCR Detection System. All primers used for RT-qPCR are listed in Supplementary Table 3. *GUSB* was used as a housekeeping gene.

**SDS-PAGE followed by western blot**. Cells were washed twice with ice-cold PBS and resuspended in RIPA buffer (0.1% SDS, 1% Igepal CA-630, 150 mM NaCl, 0.5% Sodium Deoxycholate, 20 mM EDTA) supplemented with Complete™, EDTA-free Protease Inhibitor Cocktail (1 tablet/50 ml buffer) (Sigma-Aldrich, #11873580001). Samples were kept on ice for 10 min followed by centrifugation at 16,000 g for 10 min at 4 °C. Protein concentration of the supernatant was determined with Pierce™ BCA Protein Assay Kit (Thermo Fisher Scientific, #23225). For western blotting, samples were resolved on Novex™ NuPAGE™ 4–12% Bis-Tris protein gels (Invitrogen, #NP0321BOX) and transferred to Immobilon-FL PVDF membrane (Merck, #IPFL00010). The membrane was blocked for 1 h in 4% skim milk in PBS followed by incubation overnight at 4 °C with primary antibodies: RαSMCHD1 (1:1000, Abcam #ab176731), RαLRIF1 (1:1000, Proteintech #26115-1-AP) and Mα-αTubulin (1:4000, Sigma-Aldrich #T6199). The next day, membranes were washed twice with PBS-T (0.01% Tween 20) and incubated with following secondary antibodies: IRDye® 800CW goat anti-rabbit IgG (1:10,000, Li-cor #P/N

925-32211) and IRDye® 680CW donkey anti-mouse IgG (1:10,000, #P/N 925-68072) for 1 h at room temperature. Membranes were washed twice with PBS-T prior scanning on Odyssey® CLx Imaging System (Li-cor).

**DR1 and FAS-PAS region methylation analysis by bisulfite PCR followed by TOPO-TA subcloning**. Bisulfite conversion of genomic DNA was carried out using the EZ DNA Methylation-Lightning kit (Zymo Research, #D5030) according to the manufacturer's protocol. Converted DNA was used to amplify the DR1 region using FastStart™ Taq DNA polymerase (Sigma-Aldrich, #12032902001) with the following primers: 5'-TCGTCGGCAGCGTCAGATGTGTATAAGAGACAGGGG TTGAGGGTTGGGTTTATA-3' and 5'-GTCTCGTGGGCTCGGAGATGTGTAT AAGAGACAGACAAAACTCAACCTAAAAATATAC-3'. Converted DNA was amplified at the FAS-PAS region with AccuPrime™ Taq high fidelity DNA Polymerase (Thermo Fisher Scientific, #12346086) with the following primers: 5'-ATA GGGGAGGGGGTATTTTA-3' and 5'-ACRATCAAAAACATACCTCTATCTA-3'.

PCR products were resolved on 2% TBE agarose gel, gel extracted with NucleoSpin Gel & PCR Clean-up kit (Bioke, #740609) and subcloned into the TOPO-TA vector (Thermo Fisher Scientific, #45-064-1) according to manufacturer's protocol. Plasmids were isolated from independent bacterial colonies and sent for Sanger sequencing (Macrogen). BiQ Analyzer software was used for the methylation analysis.

**Chromatin immunoprecipitation (ChIP)**. Cells were crosslinked for 10 min at room temperature with formaldehyde of 1% final concentration. The reaction was quenched by adding glycine to 125 mM final concentration. Cells were washed twice with PBS containing 0.5 mM PMSF (Sigma-Aldrich, #93482), collected by scraping and spun at 500 g for 10 min at 4 °C. Cell pellets were resuspended in the ice-cold ChIP buffer (1.5 ml lysis buffer/$10 \times 10^6$ cells) (150 mM NaCl, 50 mM Tris-HCl pH 7.5, 5 mM EDTA, 0.5% Igepal CA-630, 1% Triton X-100) supplemented with cOmplete™ Protease Inhibitor Cocktail table (Sigma-Aldrich, #11697498001). After 10 min incubation on ice, samples were spun down at 8000 g for 2 min at 4 °C. The nuclear pellets were again resuspended in ChIP buffer, incubated for 5 min on ice and followed by another round of centrifugation. Final nuclear pellets were resuspended in the ChIP buffer and sonicated at the highest power output for 25 cycles (1 cycle: 30 sec ON/30 sec OFF) using a Bioruptor instrument (Diagenode). For ChIP, chromatin was first pre-cleared with BSA-blocked protein A Sepharose beads (GE Healthcare, #17-5280-21) by rotating for 30–60 min at 4 °C. For histone ChIP, 3 μg of chromatin was used and for SMCHD1 and LRIF1 ChIP, 30 μg of chromatin was used in a final volume of 500 μl. 50 μl (10%) of each chromatin was kept as the input sample for later normalization. ChIP was carried out by rotation at 4 °C with following primary antibodies: RαSMCHD1 (Abcam, #ab31865), RαLRIF1 (Merck, #ABE1008), RαH3 (Abcam, ab1791), RαH3K4me2 (Active Motif, #39141), RαH3K9me3 (Active Motif, #39161) or RαH3K27me3 (Merck, #07-449). As a negative control, isotype rabbit polyclonal IgG was used (Abcam, #ab37415). The second day, 20 μl of BSA-blocked protein A Sepharose beads were added to all samples and incubated for 2 h at 4 °C while rotating. Afterwards, beads were washed as follows: once with low salt wash buffer (1% Triton X-100, 0.1% SDS, 2 mM EDTA, 20 mM Tris-HCl, 150 mM NaCl), high salt wash buffer (1% Triton X-100, 0.1% SDS, 2 mM EDTA, 20 mM Tris-HCl, 500 mM NaCl), LiCl wash buffer (250 mM LiCl, 1% Igepal CA-630, 1% sodium deoxycholate, 1 mM EDTA, 10 mM Tris-HCl) and twice with TE wash buffer (10 mM Tris-HCl, 1 mM EDTA). For DNA extraction, 10% (w/v) of Chelex 100 resin was added to the beads and boiled at 95 °C for 10 min while shaking. Supernatant was used for qPCR analysis. Primers used to quantify immunopurified DNA can be found in Supplementary Table 4.

**Immunofluorescent staining**. Cells were grown on collagen-coated glass-bottom 96-well plates (Greiner Bio-One, #655892) and differentiated for 2–3 days prior staining. Cells were fixed with 2% paraformaldehyde diluted in 1x PBS for 7 min at RT, followed by permeabilization with 1% Triton X-100 diluted in 1x PBS for 10 min at RT. The primary antibody against MYH1E (MF20, Hybridoma Bank, Iowa University) was diluted 1:200 in 1x PBS and incubated with the fixed cells over-night at 4 °C. Next day, primary antibody was washed away with 1xPBS and cells were incubated with the secondary antibody (1:500 dilution in 1xPBS) goat-anti-mouse Alexa 594 (Thermo Fisher Scientific, # A21203). Cells were washed with 1x PBS containing 1:1000 dilution of DAPI (Sigma-Aldrich, #268298) for nuclei visualization. Stained cells were imaged with Thermo Cellomics ArrayScan VTI HCS Reader and 100 images per cell line were taken at 20× magnification. Images were analyzed using CellProfiler Software (v2.1.1) with a custom made analysis pipeline. In short, nuclei were segmented based on DAPI staining and individual nuclei were identified based on shape and size. Myotubes were segmented based on MYH1E staining and used as mask overlay to discriminate myotube nuclei from myoblast nuclei. Fusion index was calculated as the percentage of myotube nuclei as compared to the total number of nuclei per image.

**Poly-A RNA-seq and data analysis**. Total RNA was isolated as described above and poly-A RNA-seq was outsourced to GenomeScan B.V.. Sequencing libraries were prepared with NEBNext® Ultra™ II RNA Library Prep Kit for Illumina® kit (New England Biolabs, #E7775) according to the manufacturer's manual. Samples were

sequenced as 150 bp paired-end on a NovaSeq6000 instrument. Quality assessment of the raw sequencing reads was done using FastQC v0.11.6. Adapters were removed by TrimGalore v0.4.5 with option–paired. The remaining quality-filtered reads were aligned to the human reference genome (version hg38) with the corresponding annotation file from Ensemble using the STAR aligner v2.5.1. Read count table was obtained with HTSeq-count v0.9.1 using the GENCODE V29 annotation with the option "–stranded no". The differential expression statistical analysis was done with DESeq2 v1.24.0 (R package) with default settings. The final list of differentially expressed genes contains genes for which the adjusted p-value (Benjamini-Hochberg correction) is <0.05. RNA-seq plots were generated with ggplot2 v3.3.3 (R package). Raw sequencing files have been under GEO accession number GSE185511.

**Statistics and reproducibility**. We performed each experiment on at least three biological replicates and statisctically analyzed data by GraphPad Prism (GraphPad Software, La Jolla, CA, USA). Figure legends describe specifics of statistical analysis of each experiment.

RNA-seq data was statistically analysed with DESeq2 v1.24.0 (R package) with default settings.

**Reporting summary**. Further information on research design is available in the Nature Portfolio Reporting Summary linked to this article.

## Data availability

RNA seq data was deposited as raw sequencing files under GEO accession number GSE185511.

Datasets generated during the study are all presented in figures, tables and supplementary information. Supplemental data 2 file includes all numerical source data corresponding to figures of the manuscript. Supplementary Figures 7 and 8 show unedited western blots. All other data is available upon a reasonable request from the corresponding author.

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

## Acknowledgements

We thank members of the Van der Maarel lab for all their helpful suggestions and we are thankful to LUMC Flow cytometry Core Facility for their technical support. We also thank the funding agencies who supported the study. This study was supported by grants from the US National Institute of Arthritis and Musculoskeletal and Skin Diseases (R01AR066248) and the Prinses Beatrix Spierfonds (W.OP14–01; W.OR15–26). D.S., IW, A.v.d.H., J.B. and S.M.v.d.M. are members of the European Reference Network for Rare Neuromuscular Diseases [ERN EURO-NMD].

## Author contributions

D.Š., S.J.T., L.D., J.B. and S.M.v.d.M. conceived the study. D.Š., A.M.T., I.W., J.B. performed experiments. D.Š., A.v.d.H. and J.B. analyzed data. D.Š., J.B. and S.M.v.d.M. wrote the manuscript.

## Competing interests

The authors declare no competing interests.
