## [Peer Review File · Communications Biology]

Reviewers' comments:

Reviewer #1 (Remarks to the Author):

Review of Sikrova et al. Communications Biology COMMSBIO-22-2577.

In this manuscript the authors investigate the role of SMCHD1 and LRIF1 invalidation in somatic cells on DUX4 expression, D4Z4 DNA methylation and chromatin marks at the D4Z4 repeat. They further investigate the inter-dependence of LRIF and SMCHD1 on binding to D4Z4 and regulation of D4Z4 chromatin features.

The work is mostly based on cellular models (patient's and healthy donor's derived myoblasts or fibroblasts).

As published by others in different cellular models, figure 1 and 2 (and result section up to line 167) mostly show that somatic invalidation of SMCHD1 does not change D4Z4 methylation profile, or D4Z4 chromatin features with a moderate impact on DUX4 expression. Given that all data have already been published by others, these two figures do not bring anything novel and might be presented as supplementary figure for characterization of the cellular model in use.

The main novelty of the work relates to the impact of LRIF1 on SMCHD1 binding (figure 3A). Regarding this point, the authors also need to indicate that H3K9me3 and DNA methylation are insufficient for LRIF1 recruitment to D4Z4 (end of line 191).

Regarding DNA methylation, a number of reports consider that bisulfite-specific primers mapping the 3' end of the DUX4 transcript and the A-type region are more specific to monitor D4Z4-associated DNA methylation changes. The authors might consider monitoring chromatin changes in this region as well.

Figure 3C-E. None of the results presented show a significant difference between conditions. This again strongly questions the relevance of the model. This question remains of interest in the field, for understanding FSHD pathogenesis or finding a cure but requires clear conclusions, beyond data description.

Figure 4A, lower panel, the authors need to acknowledge in the text that the observed effect on D4Z4 binding is quite moderate.

As no change in LRIF1 binding is observed in SMCHD1 corrected clones, this means that as also described in the past (ref 38), somatic invalidation of either SMCHD1 or LRIF1 has no impact on D4Z4 chromatin features and that all modifications observed in human tissues are linked to the germline mutation of related genes.

Line 345. Dion et al. showed that DNMT3B KO in somatic cells does not alter D4Z4 methylation level.

The overall conclusion of part of this work is that LRIF is involved in germline D4Z4 methylation but not required for maintenance of chromatin marks (histone repressive marks and DNA methylation) once the methylation profile is established, as shown in reference 38.

This overall indicates that modulating SMCHD1 or LRIF1 level in somatic cells might not be efficient to cure FSHD.

The authors need to indicate throughout the text that SMCHD1 has been implicated in X inactivation in mouse cells as it has not been proven in human cells.

The recent articles on the role of LRif1 in maternal transmission should be mentioned in the text.

Reviewer #2 (Remarks to the Author):

In their manuscript Šikrová et al, generated immortalized myoblasts KO for SMCHD1 or LRIF1, which are involved in the regulation of DUX4 expression in FSHD. Surprisingly, they found that removal of the factors has a minor impact on the expression of DUX4 and on the epigenetic status of the FSHD locus. Similarly, performing SMCHD1 gene correction in FSHD2 cells does not change the epigenetic status of the FSHD locus despite it is associated to DUX4 silencing. Notably, the authors found that SMCHD1 and LRIF1 regulate the expression of the LRIF1 gene. Contrary to the FSHD locus, clear epigenetic alterations of the LRIF1 locus are associated to SMCHD1 or LRIF1 modulation. Hence, the effects on the expression of DUX4 and the epigenetic make-up of the FSHD locus appear very different if the modulation of SMCHD1 or LRIF1 takes place developmentally or in somatic cells, and very different from the LRIF1 locus.

The manuscript is very intriguing, confirms and extend previous data (PMID: 30698748), and provides results relevant for both understanding FSHD pathogenesis and develop possible treatments. Nevertheless, the following aspects should be addressed by the authors to further support their findings.

1. How does the level of DUX4 expression obtained upon SMCHD1 or LRIF1 acute knockdown compare to that of stable knockout? Is it possible that the reactivation of DUX4 upon CRISPR is relatively low because KO cells adapt and compensate to the prolonged absence of SMCHD1 or LRIF1?
2. The authors have previously reported that SMCHD1 knockdown causes a significant increase in the recruitment of PRC2 factors and an increase in the deposition of H3K27me3 (Epigenetics, 10:12, 1133-1142). Connected to my previous comment, could it be that the failure to reproduce this result in SMCHD1 KO cells is due to cell adaptation to the chronic SMCHD1 absence?
3. Was the experiment of Figure 4C performed in proliferating myoblasts or differentiated myotubes? The effect on DUX4 expression would be better appreciated in differentiated cells. This will also allow a better comparison with the results of Figure 1B.

4. The primers used by the authors for their DNA methylation and ChIP experiments are not specific for the last, more telomeric D4Z4 unit, which is the one responsible for the disease. Their ChIP-qPCR and DNA methylation studies report the average levels of epigenetic marks throughout the D4Z4 arrays on both chromosome 4 and 10. Could it be that modulation of SMCHD1/LRIF1 levels in somatic cells mostly affects the last D4Z4 unit and this more subtle effect, which nevertheless would be sufficient to modulate DUX4 expression, is masked by the non 4qA-specific readout that they are using? Can the authors analyze selectively the epigenetic status of the last D4Z4 unit in 4qA alleles?

5. How is chromatin accessibility and 4q35 loop domain organization upon SMCHD1 or LRIF1 modulation in somatic cells? How does it compare to that of FSHD1 or FSHD2 muscle cells?

Reviewer #3 (Remarks to the Author):

This submission of Šikrová proposes that SMCHD1 and LRIF1 form an auxiliary layer of DUX4 repression on top of known D4Z4 repressive mechanisms, even at epigenetically compromised D4Z4 repeats. This work seeks to connect known roles of SMCHD1 and LRIF1 in regulating D4Z4 repression. The authors use ChIP-qPCR to test the chromatin changes in somatic loss-of-function mutant derivatives of either SMCHD1- or LRIF1-myoblasts. In addition, they claim that SMCHD1 and LRIF1 serve as negatively transcriptional regulators of the LRIF1 gene by binding to the LRIF1 promoter in somatic cells. Some of these findings confirm previous data obtained with somatic knockouts of SMCHD1 in a cancer cell line. The story is of relevance to the field, but I do not feel that the data is completely satisfactory to support the claimed conclusions for some of the results. There are some concerns that should be addressed.

Major concerns:

1. Figure 3A. The result show that LRIF1 protein binding at the D4Z4 is from the LRIF1S isoform to a large extent, suggesting LRIF1S serves a function at this region. if this is true, then how to explain that LRIF1 homozygous frameshift mutation leads to the loss of the long LRIF1 isoform (LRIF1L), while the expression of the short isoform (LRIF1S) is not affected in FSHD2 disease (Neurology. 2020)?

2. Figure 3A. Considering the decreased degree of ChIP, it seems that LRIF1 binding in SMCHD1 KO cells is the same as in the LRIF1L+S KO. Theoretically, there is no LRIF1 protein in LRIF1L+S KO cells; if this is the baseline of the ChIP-qPCR, SMCHD1 should be essential for LRIF1 binding at D4Z4 based on the results. Maybe it is asking too much, but it would be good to further demonstrate how and to what extent SMCHD1 is involved in LRIF1 binding at the D4Z4 region. This would be important for this study.

3. Figure S3. How to explain the differential enrichment patterns of SMCHD1 and LRIF1 in different regions of D4Z4 in the absence of DNA methylation? For example, decreased binding in the DR1 region versus increased binding in the HOX region? It is a little confusing that the hypomethylated HOX region

is associated with increased SMCHD1 enrichment.

4. Based on Figure 4A, the authors claimed that both SMCHD1 and LRIF1 recruitment to D4Z4 is sensitive to chromatin changes associated with DNA hypomethylation (the DR1 region in D4Z4 showed the most obvious decreased binding of LRIF1, and SMCHD1 also showed the decreased pattern but not significant), and on the other hand, histone modification (including H3 K9me3, K4me2 and K27me3) are only altered at the Q region (Figure S4). So, how do we explain that the binding changing site (DR1 region) of SMCHD1 and LRIF1 is not the same as the histone modification alteration locus (Q region)?

5. In this study, the authors did much ChIP-qPCR work to identify the enrichments for SMCHD1, LRIF1 and histone modifications at the D4Z4 locus in somatic loss-of-function cells of either SMCHD1- or LRIF1 (from unrelated control Individuals), such as in Figure 2D, 3A, 5D. If the same ChIP-PCR experiment can be performed in FSHD2 patient myoblasts as a control under the same experimental conditions, this approach would be much better and more interesting to provide evidence of how much the enrichment differences in the SMCHD1-KO, LRIF1-KO and FSHD2 patients are related, and how much of these differences may be involved in the potential pathogenesis of FSHD2.

Minor concerns:

1. Row 105: "This suggests that SMCHD1 is not required for DNA methylation or H3K9me3 maintenance" as mentioned by the authors is not exactly correct. In fact, the impact of SMCHD1 on DNA methylation and H3K9me3 maintenance is still a mystery. Here, SMCHD1 is not required for DNA methylation or H3K9me3 maintenance in a particular cell line, such as HCT116, which would be more exact. Based on current knowledge about SMCHD1 in different cell lines or myoblasts from FSHD2 patients, the function and exact mechanism of SMCHD1 is still not clear, but one thing is confirmed: that the functional mechanism should be different in different cell types, so, the conclusion claimed in this study should be limited to the particular cell lines used in this paper.

2. Row 229-230, and Row 250. The authors claimed that somatic loss of DNA methylation in HCT116 DKO cells leads to 5' to 3' redistribution of SMCHD1 along the D4Z4 unit, as well as LRIF1. From Figure S3, "redistribution" is not so convincing to me, because the fold change is not so remarkable, please correct it in a more exact way.

Response to Reviewer #1

Sikrova et al. Communications Biology COMMSBIO-22-2577.

We thank Reviewer #1 for the valuable comments and share our response point by point in this document.

1) As published by others in different cellular models, figure 1 and 2 (and result section up to line 167) mostly show that somatic inactivation of SMCHD1 does not change D4Z4 methylation profile, or D4Z4 chromatin features with a moderate impact on DUX4 expression. Given that all data have already been published by others, these two figures do not bring anything novel and might be presented as supplementary figure for characterization of the cellular model in use.

Response

We thank the Reviewer for this valuable comment. While we acknowledge the earlier reports on the somatic inactivation of SMCHD1, we argue that our work is the first to characterize the transcriptional and chromatin consequences of somatic LRIF1S or LRIF1S+L KO in myogenic cell lines in comparison to SMCHD1. Furthermore, this study also highlights that transcriptional derepression of DUX4 is much milder in somatic KOs than in FSHD cells.

2) The main novelty of the work relates to the impact of LRIF1 on SMCHD1 binding (figure 3A). Regarding this point, the authors also need to indicate that H3K9me3 and DNA methylation are insufficient for LRIF1 recruitment to D4Z4 (end of line 191).

Response

We extended our conclusions of this experiment also to DNA methylation. The new sentence now reads (194-196): Since H3K9me3 and DNA methylation levels at D4Z4 were not reduced in SMCHD1^{KO} cells (Figure 2D), this implies that H3K9me3 and DNA methylation are not sufficient for LRIF1 recruitment to D4Z4.

3) Regarding DNA methylation, a number of reports consider that bisulfite-specific primers mapping the 3' end of the DUX4 transcript and the A-type region are more specific to monitor D4Z4-associated DNA methylation changes. The authors might consider monitoring chromatin changes in this region as well.

Response

Initially, we monitored D4Z4 DNA methylation of the DR1 region, proximal to the *DUX4* promoter, because this region loses the most DNA methylation in FSDH2 compared to control (Hartweck et al. 2013, Ref48 in the manuscript). In response to the reviewer's request, to monitor DNA methylation distal to *DUX4*, we performed bisulfite sequencing of the FAS-PAS region and incorporated the data into Figure 2B and Suppl. Figure 2C and adjusted the text lines 154 to 160. We also included an extra schematic (Suppl. Figure 2A) to explain the difference between the two regions and their presence on 4q and 10q alleles. The schematic also contains the composition of 4q/10q alleles in the control cell lines used in this study.

4) Figure 3C-E. None of the results presented show a significant difference between conditions. This again strongly questions the relevance of the model. This question remains of interest in the field, for understanding FSHD pathogenesis or finding a cure but requires clear conclusions, beyond data

description.

Response

We thank the Reviewer for the critical comment and argue that Figure 3C, E panels show relevant data. This model represents two conditions in the same genetic background where the SMCHD1 level is modulated. Higher SMCHD1 levels lead to increased enrichment of SMCHD1 at D4Z4, associated with decreased *DUX4* expression, remarkably without any detectable change in the chromatin landscape of an FSHD2 chromatin. It corroborates the observation of earlier publications (Balog et al. 2015, Hiramuki et al. 2018, Goossens et al. 2018) that modulating SMCHD1 enrichment to D4Z4 might be an effective therapeutic approach in FSHD as it effectively silences *DUX4*.

5) Figure 4A, lower panel, the authors need to acknowledge in the text that the observed effect on D4Z4 binding is quite moderate.

Response

We agree with the Reviewer that the observed reduction in SMCHD1 and LRIF1 enrichment presented in Figure 4A is modest but resembles SMCHD1 mutation/FSHD2 situation. We adjusted the text lines 264 and 265. It now reads: SMCHD1 and LRIF1 enrichment levels in DNMT3B^{het} and DNMT3B^{bi} fibroblasts were similar to enrichment levels measured in SMCHD1^{het} (FSHD2) fibroblasts.

6) As no change in LRIF1 binding is observed in SMCHD1 corrected clones, this means that as also described in the past (ref 38), somatic invalidation of either SMCHD1 or LRIF1 has no impact on D4Z4 chromatin features and that all modifications observed in human tissues are linked to the germline mutation of related genes.

Response

We agree with the Reviewer and believe our manuscript concurs with this statement.

7) Line 345. Dion et al. showed that DNMT3B KO in somatic cells does not alter D4Z4 methylation level.

Response

We agree with the Reviewer that in HCT116 cancer cell line knocking out DNMT3B did not result in D4Z4 hypomethylation, however, this has not been corroborated in somatic skeletal muscle cells yet. Therefore, we are now more specific in the sentence of concern. The sentence now reads (line 356): However, whether catalytically active or inactive DNMT3B isoforms have a physiologically relevant function in D4Z4 repression in skeletal muscle cells after methylation patterns have been established in early embryogenesis, remains to be addressed.

8) The overall conclusion of part of this work is that LRIF is involved in germline D4Z4 methylation but not required for maintenance of chromatin marks (histone repressive marks and DNA methylation) once the methylation profile is established, as shown in reference 38. This overall indicates that modulating SMCHD1 or LRIF1 level in somatic cells might not be efficient to cure FSHD.

Response

We agree with the Reviewer's interpretation of the epigenetic establishment and maintenance function of SMCHD1 and LRIF1. But we disagree with the conclusion about the therapeutic potential of SMCHD1 or LRIF1. The following publications consistently show that increasing SMCHD1 levels in DUX4 expressing myogenic cells leads to decreased DUX4 expression: Balog et al. 2015, Hiramuki et al. 2018, Goossens et al. 2018.

9) The authors need to indicate throughout the text that SMCHD1 has been implicated in X inactivation in mouse cells, as it has not been proven in human cells.

Response

We thank the Reviewer for this comment and adapted the manuscript's text in lines 70 and 377-378.

10) The recent articles on the role of LRif1 in maternal transmission should be mentioned in the text.

Response

We thank the Reviewer for reminding us about this observation, and we adapted the manuscript by mentioning it in lines 71 and 97-98.

Response to Reviewer #2

Sikrova et al. Communications Biology COMMSBIO-22-2577.

We thank Reviewer #2 for the valuable comments and share our response point by point in this document.

1. How does the level of DUX4 expression obtained upon SMCHD1 or LRIF1 acute knockdown compare to that of stable knockout? Is it possible that the reactivation of DUX4 upon CRISPR is relatively low because KO cells adapt and compensate to the prolonged absence of SMCHD1 or LRIF1?

Response

We thank the Reviewer for the comment. Our earlier publication (Sikrova et al., 2020) describes the transcriptional changes after short-time depletion of LRIF1L (Figure 3A) in control immortalized myoblasts. We observed that the complete depletion of LRIF1L was unsuccessful with siRNAs, and the residual amount of protein could still modulate DUX4 chromatin. This observation motivated us to generate the KO clones.

Figure 1B shows DUX4 and selected DUX4 target expression levels of the different KO clones next to two FSHD2 samples, and we comment on it in lines 137-139. Figure 1B shows that DUX4 levels measured in any of the KO clones are approximately 50 times lower than FSHD2 samples. We did not perform acute knockdown of SMCHD1 or LRIF1 in the parental cell lines, so we cannot compare acute depletion with the KO situation. We agree with the Reviewer that adaptation of the chromatin landscape could have happened during the generation of the KO clones, which might influence the transcriptional activity of the locus. We have therefore included this limitation in our discussion (lines 369-371).

2. The authors have previously reported that SMCHD1 knockdown causes a significant increase in the recruitment of PRC2 factors and an increase in the deposition of H3K27me3 (Epigenetics, 10:12, 1133-1142). Connected to my previous comment, could it be that the failure to reproduce this result in SMCHD1 KO cells is due to cell adaptation to the chronic SMCHD1 absence?

Response

We thank the Reviewer for the comment. We agree with the Reviewer that adaptation of the chromatin landscape could have happened during the generation of the KO clones and have mentioned this limitation to the discussion (lines 369-371). Our earlier publication (Balog et al., 2016) presents data generated on primary myoblasts and the short-term depletion of SMCHD1 where possible adaptation mechanisms could not occur.

3. Was the experiment of Figure 4C performed in proliferating myoblasts or differentiated myotubes? The effect on DUX4 expression would be better appreciated in differentiated cells. This will also allow a better comparison with the results of Figure 1B.

Response

We agree with the Reviewer that DUX4 expression correlates with differentiation. Since we often observe differences in differentiation between siRNA treated samples and since SMCHD1 KO clones show an enhanced fusion index (Suppl. Figure 1B), we decided to use proliferating cells to avoid differentiation differences between KO conditions which might interfere with DUX4 expression levels. We modified the text at line 273 to clearly state the cell condition used in the investigation.

4. The primers used by the authors for their DNA methylation and ChIP experiments are not specific for the last, more telomeric D4Z4 unit, which is the one responsible for the disease. Their ChIP-qPCR and DNA methylation studies report the average levels of epigenetic marks throughout the D4Z4 arrays on both chromosome 4 and 10. Could it be that modulation of SMCHD1/LRIF1 levels in somatic cells mostly affects the last D4Z4 unit and this more subtle effect, which nevertheless would be sufficient to modulate DUX4 expression, is masked by the non 4qA-specific readout that they are using? Can the authors analyze selectively the epigenetic status of the last D4Z4 unit in 4qA alleles?

Response

We thank the Reviewer for the suggestion. We now examined an additional region for DNA methylation known as FasPAS using previously established primers (Calandra P. et al., 2016). FasPAS region lies immediately distal to the last repeat unit at 4qA haplotype (Suppl. Figure 2A) and should be thus a single copy region which reports methylation level specifically of *DUX4* expressing allele in our control lines and their KO derivatives. We incorporated the data into Figure 2B and Suppl. Figure 2C and adjusted the text lines 154-156.

5. How is chromatin accessibility and 4q35 loop domain organization upon SMCHD1 or LRIF1 modulation in somatic cells? How does it compare to that of FSHD1 or FSHD2 muscle cells?

Response

We agree with the Reviewer that it would be interesting to study the role of SMCHD1 and LRIF1 in the topological organization of 4q35. Topological studies of D4Z4 are challenging because of the repetitive nature of the genomic region and the presence of many D4Z4 like sequences scattered throughout the genome. We believe this is beyond the scope of this manuscript and warrants further investigations.

Response to Reviewer #3

Sikrova et al. Communications Biology COMMSBIO-22-2577.

We thank Reviewer #3 for the valuable comments and share our response point by point in this document.

Major concerns:

1. Figure 3A. The result show that LRIF1 protein binding at the D4Z4 is from the LRIF1S isoform to a large extent, suggesting LRIF1S serves a function at this region. if this is true, then how to explain that LRIF1 homozygous frameshift mutation leads to the loss of the long LRIF1 isoform (LRIF1L), while the expression of the short isoform (LRIF1S) is not affected in FSHD2 disease (Neurology. 2020)?

Response

The Reviewer is correct that in Figure 3A LRIF1 enrichment at D4Z4 in LRIF1L^{KO} and control clones is similar which might suggest a specific function for LRIF1S at D4Z4 chromatin. What exactly this function is, and whether this overlaps with LRIF1L function, we do not know. But we do know that the presence of LRIF1S is insufficient to suppress D4Z4 transcriptionally as supported by our current data in the somatic LRIF1L KO clones and by patient data described in Sikrova et al., 2020.

The expression of LRIF1S in the reported patient is unaffected because the frameshifting mutation is present in exon 2, which is specific to the long isoform of LRIF1 and thus does not affect the expression of the short isoform.

2. Figure 3A. Considering the decreased degree of CHIP, it seems that LRIF1 binding in SMCHD1 KO cells is the same as in the LRIF1L+S KO. Theoretically, there is no LRIF1 protein in LRIF1L+S KO cells; if this is the baseline of the CHIP-qPCR, SMCHD1 should be essential for LRIF1 binding at D4Z4 based on the results. Maybe it is asking too much, but it would be good to further demonstrate how and to what extent SMCHD1 is involved in LRIF1 binding at the D4Z4 region. This would be important for this study.

Response

We agree with the Reviewer that our current manuscript does not fully identify the molecular mechanism of LRIF1 binding to D4Z4. Our work shows that SMCHD1 is necessary for LRIF1L recruitment to D4Z4 chromatin as in somatic SMCHD1 KO cells, LRIF1 does not bind to D4Z4 following the argument of the reviewer (Figure 3A). But SMCHD1 is insufficient to recruit LRIF1 to D4Z4 when its chromatin structure is already compromised (Figure 3D) as increasing SMCHD1 does not lead to increased enrichment of LRIF1 at D4Z4. Further work on other chromatin components can shed light on the detailed mechanism. In our discussion we mention several factors that might cooperate in LRIF1 recruitment to D4Z4 but our studies so far have failed to unequivocally identify the detailed mechanism of LRIF1 recruitment to D4Z4.

3. Figure S3. How to explain the differential enrichment patterns of SMCHD1 and LRIF1 in different regions of D4Z4 in the absence of DNA methylation? For example, decreased binding in the DR1 region

versus increased binding in the HOX region? It is a little confusing that the hypomethylated HOX region is associated with increased SMCHD1 enrichment.

Response

We agree with the Reviewer that it is puzzling that two different regions of D4Z4 recruit SMCHD1 and LRIF1 with different affinities in hypomethylation conditions; however, other researchers observed the same phenomenon (Dion et al., 2019). As pointed out in our discussion, we feel it is important to emphasize that one of the differences might be related to the single copy nature of the HOX region versus the multicopy array nature of the D4Z4 locus.

4. Based on Figure 4A, the authors claimed that both SMCHD1 and LRIF1 recruitment to D4Z4 is sensitive to chromatin changes associated with DNA hypomethylation (the DR1 region in D4Z4 showed the most obvious decreased binding of LRIF1, and SMCHD1 also showed the decreased pattern but not significant), and on the other hand, histone modification (including H3 K9me3, K4me2 and K27me3) are only altered at the Q region (Figure S4). So, how do we explain that the binding changing site (DR1 region) of SMCHD1 and LRIF1 is not the same as the histone modification alteration locus (Q region)?

Response

We thank the Reviewer for this observation. Figure 4A shows SMCHD1 and LRIF1 enrichment at three different D4Z4 regions, while Suppl Figure 4B presents the histone profile of only one of the D4Z4 regions, closest to the promoter region of DUX4. Since we do not have data on the chromatin profile of DR1 and HOX regions, we cannot be sure how it correlates with SMCHD1 and LRIF1 binding.

Several publications describe SMCHD1 being involved in the topological organization of chromatin; changes in SMCHD1 (and probably LRIF1) binding affect the histone landscape of distant chromatin regions as described for the Xi and the Hox clusters in mice (Jansz et al., 2018, ref30).

5. In this study, the authors did much ChIP-qPCR work to identify the enrichments for SMCHD1, LRIF1 and histone modifications at the D4Z4 locus in somatic loss-of-function cells of either SMCHD1- or LRIF1 (from unrelated control Individuals), such as in Figure 2D, 3A, 5D. If the same ChIP-PCR experiment can be performed in FSHD2 patient myoblasts as a control under the same experimental conditions, this approach would be much better and more interesting to provide evidence of how much the enrichment differences in the SMCHD1-KO, LRIF1-KO and FSHD2 patients are related, and how much of these differences may be involved in the potential pathogenesis of FSHD2.

Response

We thank the Reviewer for the comment. We performed SMCHD1 and LRIF1 ChIP-qPCR in primary control and FSHD2 myoblasts and incorporated the data into our manuscript from line 195 onwards and in Suppl Figure 3A. The data from the primary cell cultures support the data the knock out situations.

Minor concerns:

1. Row 105: *“This suggests that SMCHD1 is not required for DNA methylation or H3K9me3 maintenance” as mentioned by the authors is not exactly correct. In fact, the impact of SMCHD1 on DNA methylation and H3K9me3 maintenance is still a mystery. Here, SMCHD1 is not required for DNA methylation or H3K9me3 maintenance in a particular cell line, such as HCT116, which would be more exact. Based on current knowledge about SMCHD1 in different cell lines or myoblasts from FSHD2 patients, the function and exact mechanism of SMCHD1 is still not clear, but one thing is confirmed: that the functional mechanism should be different in different cell types, so, the conclusion claimed in this study should be limited to the particular cell lines used in this paper.*

Response

We thank the Reviewer for this remark. We changed the sentence of concern to specifically refer to HCT116 cell line, which now reads (line 106-109): On the other hand, a recent study showed that knocking out SMCHD1 in HCT116 colon carcinoma cells leads to *DUX4* derepression without changes in DNA methylation or H3K9me3 levels at D4Z4 suggesting that SMCHD1 is not required for DNA methylation or H3K9me3 maintenance in this cell line.

2. Row 229-230, and Row 250. *The authors claimed that somatic loss of DNA methylation in HCT116 DKO cells leads to 5' to 3' redistribution of SMCHD1 along the D4Z4 unit, as well as LRIF1. From Figure S3, “redistribution” is not so convincing to me, because the fold change is not so remarkable, please correct it in a more exact way.*

Response

We thank the Reviewer for this comment. Line 229-230 and 250 refer to Suppl. Figure 3 (Suppl. Figure 4 in the new version of the manuscript). Here we present an appreciable but non-significant decrease at DR1 and a significant increase at the Hox region for SMCHD1 enrichment, while LRIF1 enrichment significantly changes at these loci in the same direction as SMCHD1. We believe that therefore that the data supports our interpretation that there is a “redistribution” of SMCHD1 and LRIF1 in this cell line similarly as observed by Dion et al.

REVIEWERS' COMMENTS:

Reviewer #1 (Remarks to the Author):

The authors addressed satisfactorily my comments and performed additional experiments that sustain and strengthen their initial findings.

The manuscript can be accepted in its revised form.

Reviewer #2 (Remarks to the Author):

The authors addressed my concerns satisfactorily.

Reviewer #3 (Remarks to the Author):

My questions have been addressed. No additional concerns.